# Anatomical organization of presubicular head-direction circuits

Patricia Preston-Ferrer, Stefano Coletta, Markus Frey, Andrea Burgalossi*

Werner-Reichardt Centre for Integrative Neuroscience, Tübingen, Germany

**Abstract** Neurons coding for head-direction are crucial for spatial navigation. Here we explored the cellular basis of head-direction coding in the rat dorsal presubiculum (PreS). We found that layer2 is composed of two principal cell populations (calbindin-positive and calbindin-negative neurons) which targeted the contralateral PreS and retrosplenial cortex, respectively. Layer3 pyramidal neurons projected to the medial entorhinal cortex (MEC). By juxtacellularly recording PreS neurons in awake rats during passive-rotation, we found that head-direction responses were preferentially contributed by layer3 pyramidal cells, whose long-range axons branched within layer3 of the MEC. In contrast, layer2 neurons displayed distinct spike-shapes, were not modulated by head-direction but rhythmically-entrained by theta-oscillations. Fast-spiking interneurons showed only weak directionality and theta-rhythmicity, but were significantly modulated by angular velocity. Our data thus indicate that PreS neurons differentially contribute to head-direction coding, and point to a cell-type- and layer-specific routing of directional and non-directional information to downstream cortical targets.

## Introduction

The initial observation made by Ranck and Taube (*Ranck, 1984*; *Taube et al., 1990a*; *1990b*) that neurons in the dorsal portion of the rat presubiculum (PreS; classically referred to as 'postsubiculum') are tuned to the head-direction (HD) of the animal, represents a milestone discovery for the neural representation of direction. Together with place cells (*O'Keefe and Dostrovsky, 1971*; *O'keefe and Nadel, 1978*), grid cells (*Hafting et al., 2005*) and border cells (*Savelli et al., 2008*; *Solstad et al., 2008*; *Lever et al., 2009*), HD cells are thought to be part of an internal representation of self-location in the mammalian brain, and hence support spatial navigation and cognition (*Valerio and Taube, 2012*; *Gibson et al., 2013*).

The discovery of HD cells was followed by many years of investigation, aimed at elucidating the subcortical and cortical networks involved in the generation and processing of HD information (*Taube, 2007*; see *Yoder and Taube, 2014*; *Geva-Sagiv et al., 2015* for review). According to current views, HD signals are generated subcortically and relayed to parahippocampal cortices via dorsal thalamic nuclei (*Taube, 1995*; *Goodridge and Taube, 1997*). The PreS receives a major projection from dorsal thalamic nuclei (*Shipley and Sorensen, 1975*; *Thompson and Robertson, 1987*; *Shibata, 1993*), contains the highest proportion of sharp HD cells among parahippocampal cortices (*Boccara et al., 2010*; *Winter et al., 2015a*) and contributes a major projection to the medial entorhinal cortex (MEC) (*Caballero-Bleda and Witter, 1993*; *1994*; *Honda and Ishizuka, 2004*). Thus, the PreS represents a major gateway of HD information into the entorhinal-hippocampal circuit. Notably, HD inputs to MEC where most grid cells have been observed (*Sargolini et al., 2006*; *Boccara et al., 2010*) have recently received great attention following experimental evidence pointing to HD signals as critical contributors to grid cell firing - in line with predictions from path-integration models (*Burak and Fiete, 2006*; *McNaughton et al., 2006*; *Bush and Burgess, 2014*). HD inputs to entorhinal grid cells could be 'un-masked' by removing excitatory feedback from the

*For correspondence: andrea.
burgalossi@cin.uni-tuebingen.de

**Competing interests:** The
authors declare that no
competing interests exist.

**Reviewing editor:** Howard
Eichenbaum, Boston University,
United States

hippocampus (*Bonnevie et al., 2013*), and grid cell firing was disrupted following inactivation of HD signals (*Winter et al., 2015a*). Thus, theoretical and experimental evidence provide support for a 'HD-to-grid' transformation, and thus point to HD signals as critical components of the 'cognitive' grid-representation of space. However, despite this progress at the computational and systems level, direct anatomical evidence has been lacking. Specifically, while a previous study has indicated that HD inputs reach the MEC (*Tukker et al., 2015*), it is currently unknown how these projections are anatomically organized, and whether the morphological/electrophysiological diversity of PreS neurons (*Funahashi and Stewart, 1997*; *Simonnet et al., 2013*; *Abbasi and Kumar, 2013*) is related to in-vivo function. These represent major limitations for understanding how parahippocampal circuits are functionally organized, and how anatomically-identified circuits support spatial cognitive functions.

In the present work we address these issues by a combined anatomical and physiological approach. Specifically, we provide evidence for a layer- and cell-type specific representation of HD in the rat PreS, and resolve the anatomical organization of long-range HD inputs to the MEC.

## Results

### Cellular organization of the superficial layers of the rat PreS

We first investigated the cytoarchitectonic and cellular organization of PreS circuits. In the present study, we targeted the dorsal PreS (*Figure 1A* and *Figure 1—figure supplement 1*), and its borders could be reliably assessed by cytoarchitectonic criteria and neuroanatomical markers (*Figure 1—figure supplement 2*). The neuronal marker NeuN and calbindin revealed a prominently modular organization of PreS layer 2 (L2; *Figure 1B*; see also *Ding and Rockland, 2001*; *Honda and Ishizuka, 2004*) while layer 3 (L3) had a more homogenous appearance (NeuN staining; *Figure 1B*). In L2, calbindin immunoreactivity (*Fujise et al., 1995*) revealed two distinct principal cell populations - calbindin-positive and calbindin-negative neurons - which represented ~33% and 67% of the total neurons within this layer, respectively (n = 916 calbindin-positive out of 2793 NeuN-positive neurons; *Figure 1B*). Similarly to the organization of MEC (*Varga et al., 2010*; *Kitamura et al., 2014*; *Ray et al., 2014*; *Fuchs et al., 2016*), calbindin-positive PreS L2 neurons were also arranged in clusters, and their dendrites bundled together and formed tent-like structures in layer 1 (L1; *Figure 1B*).

The PreS is known to project to many downstream cortical and subcortical targets, with the projection to MEC representing the most prominent output (*Shipley and Sørensen, 1975*; *van Groen and Wyss, 1990b*; *Caballero-Bleda and Witter, 1993*; *1994*; *Honda and Ishizuka, 2004*; *Honda et al., 2008*). We next explored the cellular and laminar specificity of this cortical output by injecting the retrograde neuronal tracer Cholera-toxin subunit B (CTB) in MEC. In line with previous work (*Caballero-Bleda and Witter, 1993*; *Honda and Ishizuka, 2004*), we found that the PreS projection to MEC is layer-specific, since the large majority of retrogradely-labelled neurons was found in ipsilateral (and contralateral; not shown) PreS L3 (*Figure 1C,D*; 4497 total counted neurons, n = 4 brains). On the other hand, labeling in L2 was sparse, with very few neurons contributing to this pathway (*Figure 1D*), all of which were calbindin-negative (not shown).

We next sought to explore the projection targets of the two principal cell populations (calbindin-positive and calbindin-negative neurons) in PreS L2. We found that these two cell types could be differentiated according to contralateral cortical projection targets: CTB injections in PreS resulted in dense cellular labeling in contralateral PreS L2 (*Figure 1E*; see also *Figure 1—figure supplement 3A–D*; *van Groen and Wyss, 1990a*; *Honda et al., 2008*), where the majority of retrogradely-labelled neurons was calbindin-positive (~76%, *Figure 1F*; 159 total counted L2 neurons, n = 3 brains) and arranged in clusters (*Figure 1E*). The cellular specificity of this labeling pattern was reversed by tracer injections in the superficial layers of retrosplenial (RS) cortex area 29 (*Sugar et al., 2011*; *Boccara et al., 2015*; *Sugar and Witter, 2016*) whose rostral border with PreS could be reliably identified based on the differential expression of calbindin, wolframin and zinc (*Figure 1—figure supplement 2*). CTB injections centered on this area resulted in intense cellular labeling in the contralateral homotypical area (*Figure 1G* and *Figure 1—figure supplement 3E–H*). Within PreS, most retrogradely-labelled neurons were found within L2 (*Figure 1—figure supplement 3E–H*), the majority of which were calbindin-negative (~86%, *Figure 1G*; 896 total counted L2 neurons, n = 4 brains).

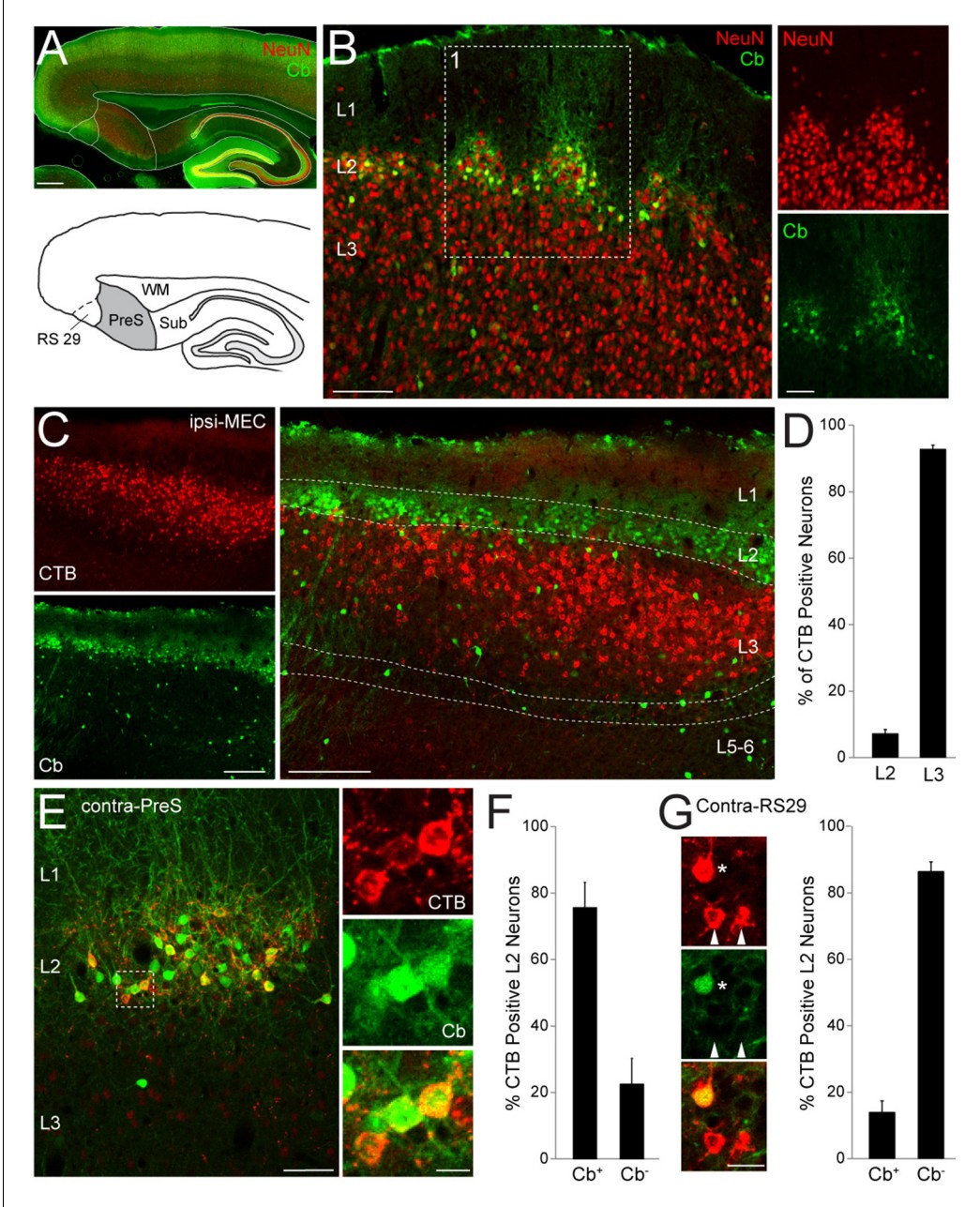

**Figure 1.** Anatomical organization and projection targets of superficial PreS neurons. (**A**) Top, parasagittal section through the dorsal PreS stained for calbindin (Cb, green) and NeuN (red). Scale bar = 500 μm. Bottom, outline of the PreS (grey) from the section shown above. RS29 indicates the subfield of RS cortex which was targeted for retrograde tracing experiments. See *Figure 1—figure supplement 1* for more details. (**B**) Superimposed staining for calbindin (green) and NeuN (red) showing the clustering of neuronal somata in L2 of PreS and the more homogeneous distribution of cells in L3. Right, close-up magnification of the single channels for panel 1 (red, NeuN; green, calbindin). Scale bars: 100 μm (left) and 50 μm (right). (**C**) Parasagittal section through PreS stained for calbindin (green) showing retrogradely-labeled neuronal somata following injection of CTB-Alexa 555 (red) in ipsilateral MEC ('ipsi-MEC'). Left panels, single channels; right panel, overlay. Scale bars: 200 μm. (**D**) Bar-graph showing the % of retrogradely-labelled (CTB-positive) neurons in L2 and L3 of PreS, following tracer injection in ipsi-MEC (as shown in **C**; 4497 total counted neurons, n = 4 brains). Error bars indicate SEM. (**E**) Parasagittal section through PreS stained for calbindin (green) showing retrogradely-labeled neuronal somata following injection of CTB-Alexa 555 (red) in contralateral PreS ('contra-PreS'). Scale bar: 50 μm. Right panel, close-up magnification of the inset shown on the left, showing three retrogradelly-labelled neurons (red) positive for the

*Figure 1 continued on next page*

*Figure 1 continued*

marker calbindin (green). Scale bar: 10 µm. (**F**) Bar-graph showing the % of calbindin-positive (Cb⁺) and calbindin-negative (Cb⁻) L2 neurons, which were retrogradely-labelled following tracer injection in contra-PreS (as shown in **E**; 159 total counted neurons, n = 3 brains). Error bars indicate SEM. (**G**) Left panels, close-up magnification PreS L2 neurons following injection of CTB-Alexa 555 (red) in contralateral RS29 and stained for calbindin (green). One calbindin-positive (asterisk) and two calbindin-negative neurons (arrowheads) are indicated. Scale bar: 10 µm. Right, bar-graph showing the % of calbindin-positive (Cb⁺) and calbindin-negative (Cb⁻) L2 neurons, which were retrogradely-labelled following tracer injection in contra-RS29 (896 total counted neurons, n = 4 brains). Error bars indicate SEM.

The following figure supplements are available for figure 1:

**Figure supplement 1.** Immunohistochemical analysis and outline of the PreS.

**Figure supplement 2.** Neuroanatomical markers outlining the rostral and caudal PreS borders.

**Figure supplement 3.** Layer distribution of retrogradely-labeled neurons in the contralateral PreS.

Altogether, these results indicate that the superficial layers of PreS (L2 and L3) can be differentiated according to cytoarchitectonic organization, cellular composition and cortical projection targets.

## Identified HD cells in the rat PreS

We next sought to investigate how HD activity, the predominant firing pattern observed among PreS neurons (*Taube et al., 1990a*; *Boccara et al., 2010*), relates to cellular and circuit heterogeneity of the superficial PreS layers. To this end, we took advantage of a head-fixed preparation (see *Zugaro et al., 2001*; *2002*; *Shinder and Taube, 2011*; *2014* for review) and recorded spiking activity from single neurons in awake rats during passive rotation. Animals were head-fixed on a rotating platform and body-centered rotations were manually performed by the experimenter (see *Video 1*). Within the same recording, animals were rotated both clockwise and counterclockwise (average number of inversions, 6.6 ± 4.7; n = 310 recordings) and average accelerations (1.3 ± 0.8 rad/s²), decelerations (−1.1 ± 0.7 rad/s²) and angular velocities (1.1 ± 0.4 rad/s) were within the physiological ranges reported by previous studies (*Blair and Sharp, 1995*; *Taube, 1995*; *Stackman and Taube, 1997*; *Shinder and Taube, 2011*). The main advantage of the head-fixed preparation - mechanical stability - allowed us to perform a large number of juxtacellular recordings from single PreS neurons (n = 310) and thus explore the cellular basis of the HD representation via juxtacellular labeling and cell identification (see below).

In line with previous studies using similar head-restraining procedures, as well as freely moving animals (*Taube et al., 1990a*; *Taube, 1995*; *Tukker et al., 2015*), sharp HD-selective responses were very common among PreS neurons and could be reliably assessed by on-line monitoring of spiking activity (see Materials and methods and *Video 1*). To quantify head-directionality of spiking responses, we computed the HD index (*Boccara et al., 2010*), while statistical significance was assessed with a shuffling test (*Boccara et al., 2010*; *Tukker et al., 2015*). Neurons were defined as HD cells if the HD index was larger than the 95th percentile of the shuffled distribution (see Materials and methods). A large proportion of PreS neurons met these criteria (186 out of 310; ~60%; see also *Boccara et al., 2010*; *Tukker et al., 2015*; *Figure 2A,B*), with the fraction 'strong' HD cells (HD index >0.8 and p<0.01; 121/310, ~39%) being within the range of previous studies from freely-behaving rodents

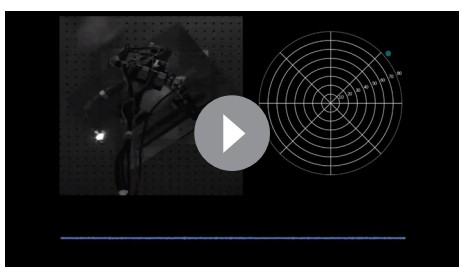

**Video 1.** Representative recording of a HD cell from the rat PreS. The video shows a recording of a PreS HD in a rat during passive rotation. A polar plot (showing total spike count as a function of HD; upper right corner) and a high-pass filtered spike trace (bottom) are displayed.

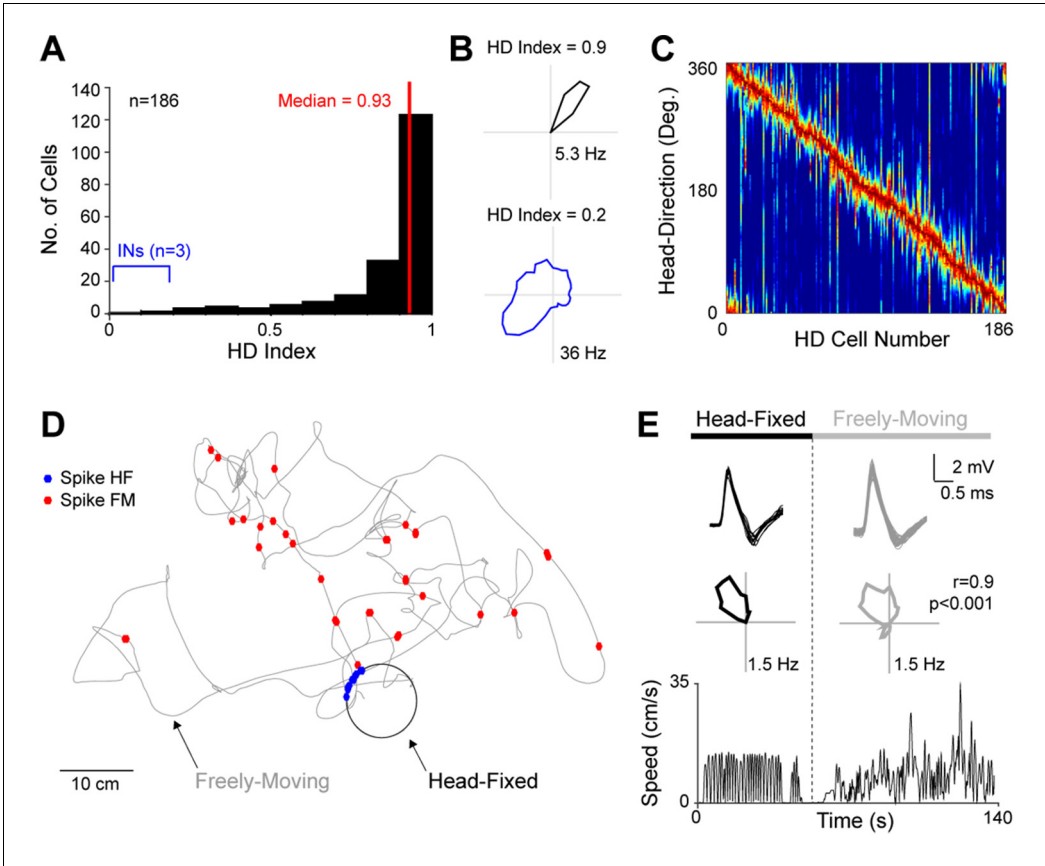

**Figure 2.** HD tuning of PreS neurons. (**A**) Histogram showing the distribution of HD Indices for all PreS neurons which met the HD criteria (n = 186; see Materials and methods). The median HD index is indicated and shown by the red line. Three recordings from putative FS INs contributed weakly-directional responses (blue; see also *Figure 2—figure supplement 1*). (**B**) Polar plots showing firing rate as a function of HD for the neuron with the highest HD index (top) and a representative FS IN (bottom; see also *Figure 2—figure supplement 1*). For the cell shown on the top panel, all spikes (n = 22) were fired within a narrow HD angle (~10 degrees). HD indices and peak firing rates are indicated. (**C**) Color-coded distribution of preferred direction for all HD cells (n = 186). Each row represents the firing rate of a single neuron (normalized relative to its peak firing rate; red), ordered by the location of their peak firing rates relative to the rat's HD. (**D**) Spike-trajectory plot for a HD cell, sequentially recorded during passive rotation ('head-fixed', HF) and free-behavior ('freely-moving', FM). The circular trajectory of the rat's head during passive rotation is indicated in black, while the rat's trajectory during free behavior in gray. Spikes fired during head-fixation and free-behavior are indicated as blue and red dots, respectively. (**E**) Superimposed spike waveforms (top), polar plots showing firing rate as a function of HD (middle) and linear velocities (bottom) computed from the passive rotation (left) and freely-moving session (right) for the recording shown in (**E**). Note the stability of the spike-shape and the similar HD tuning between the head-fixed and freely-moving session (the Pearson's correlation coefficient, p value and peak firing rates are indicated).
The following figure supplement is available for figure 2:

**Figure supplement 1.** Activity of identified and putative fast-spiking interneurons during passive rotation.

---

(*Taube et al., 1990a*; *1990b*; *Boccara et al., 2010*; *Tukker et al., 2015*). A minority of weak but statistically-significant HD responses were contributed by fast-spiking (FS) interneurons (*Figure 2A*; see below). Firing in HD cells was stable over time, as assessed by Pearson's correlation coefficient of HD tuning curves computed for the two halves of each recording session (mean correlation coefficient, 0.79 ± 0.21; n = 181 HD cells) and preferred firing directions were homogeneously distributed over a 360 degrees angle (*Figure 2C*; *Taube et al., 1990a*; *Taube, 1995*). During passive rotation, average and peak firing rates of HD cells (3.4 ± 4.2 Hz and 15.4 ± 12.6 Hz, respectively; n = 186)

were also within the range reported during free behavior (*Taube et al., 1990a*; *1990b*; *Blair and Sharp, 1996*). Thus, in line with previous work, the basic properties (e.g. distribution of preferred firing directions, HD strength, stability, average and peak firing rates) and abundance of PreS HD cells recorded under passive rotation appeared to be very similar to the ones recorded in freely-moving animals. To further confirm that bona fide HD cells can be recorded under passive rotation, in a subset of recordings (n = 4) we sequentially monitored the activity of the same HD cells during head-fixation and free-behavior. To achieve this, we used miniaturized recording equipment (*Tang et al., 2014*), which allowed us to release the rats from head-fixation while maintaining the juxtacellular recording during free movement. As shown in the representative recording in *Figure 2D*, the general tuning properties of the HD cells were very similar between passive-rotation and free behavior (*Figure 2E*; mean correlation coefficient of the HD tuning curves, 0.68 ± 0.20, p<0.05; n = 4).

Based on these results, we took advantage of this preparation for exploring the anatomical organization of HD circuits. In a subset of the recorded neurons, juxtacellular labeling was performed for obtaining cell identification. Representative recordings from identified HD cells are shown in *Figure 3*. These neurons were identified as L3 pyramidal cells, with relatively simple apical dendrites reaching L1 and basal dendrites largely confined within L3 (*Figure 3A and E*). Spikes from these identified neurons were sharply tuned to the direction the animal was facing during passive rotation (*Figure 3B and F*) with spikes occurring within a narrow directional angle (HD Index = 0.98, p=0.001; and HD Index = 0.97, p=0.004; for *Figure 3C and G*, respectively). HD firing was stable, as

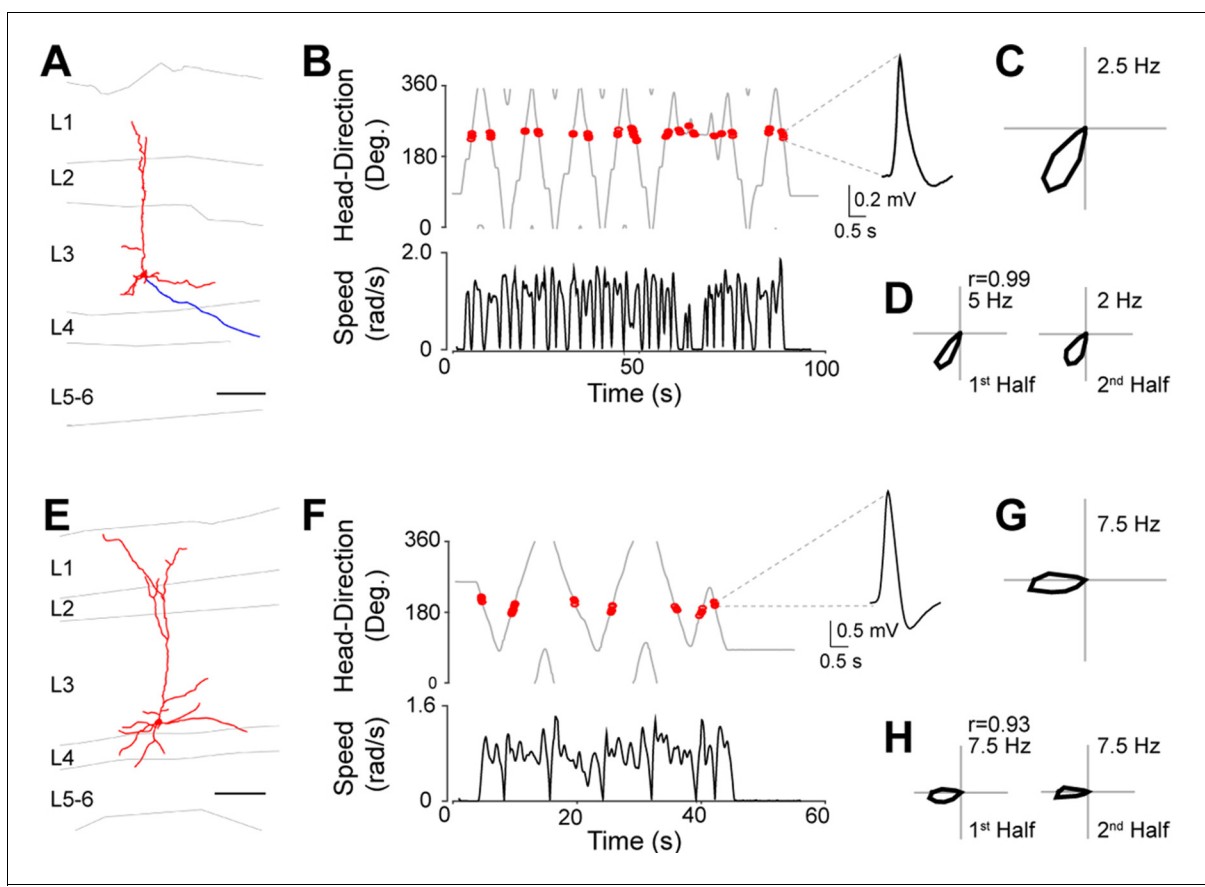

**Figure 3.** Identified HD cells in PreS layer 3. (A) Morphological reconstruction of a representative layer 3 pyramidal HD cell (dendrites, red; axon, blue). Scale bar: 100 µm. (B) Angular HD (top) and angular speed (bottom) as a function of time. Spikes (red dots) are indicated. Note the sharp tuning to HD. (C) Polar plots showing firing rate as a function of HD for the neuron in (A). Peak firing rate is indicated. (D) Polar plots showing firing rate as a function of HD computed or the two halves of the recording session for the neuron in (A). The Pearson's correlation coefficient between the two HD tuning curves and peak firing rates are indicated. (E–H) same as A–D but for another neuron. Scale bar: 100 µm.

assessed by the Pearson's correlation coefficients of HD tuning between the two halves of the recording sessions (0.99 and 0.93 for *Figure 3D and H*, respectively).

In total, we successfully labeled and recovered 54 PreS neurons (48 principal cells and 6 interneurons; see also *Figure 2—figure supplement 1*) during passive rotation. Of these, 27 (50%) were classified as HD cells. The majority of identified HD cells were located in L3 (n = 18), the rest in deep layers (L4–6; n = 9). No HD cell was recovered in L2. All principal neurons whose morphology could be assessed (see Materials and methods) were classified as pyramidal (21 out of 21 in L3; 4 out of 8 in deep layers) or multipolar neurons (4 out of 8 in deep layers). These data thus indicate that, within the superficial PreS layer, HD responses are preferentially contributed by L3 pyramidal neurons.

In our dataset, a subset of recordings could be classified as FS (n = 20) based on spike-width and firing rate criteria (*Taube et al., 1990a*; *Tukker et al., 2015*) which were confirmed by cell identification (n = 3; *Figure 2—figure supplement 1A*). In line with previous work from freely-moving rats (*Tukker et al., 2015*) we found that a minority of FS interneurons (3 out of 20) contributed weak HD responses (*Figure 2—figure supplement 1B*), which were stable between the two halves of the recording sessions (mean correlation coefficient, 0.73 ± 0.30, p<0.05; n = 3). The majority of FS interneurons (13 out of 20) were significantly modulated by angular velocity (see Materials and methods) and fired at higher rates during rotation compared to resting periods (*Figure 2—figure supplement 1C*). Theta rhythmicity was very sparse among FS interneurons (*Figure 2—figure supplement 1D,E*); yet classical 'theta-cells' were observed within the PreS (as in *Taube et al., 1990a*; *Blair and Sharp, 1996*), and one of them was identified as a paravalbumin-positive interneuron (*Figure 2—figure supplement 1E*). PreS interneurons were thus modulated by rotational movement and were on average only weakly tuned to HD and entrained by the theta rhythm.

## Long-range axonal projections of identified PreS HD cells

We next sought to explore the long-range organization of HD circuits within parahippocampal cortices. We thus performed a subset of experiments, where animals were sacrificed ~4 to 12 hrs following juxtacellular labeling to ensure long-range filling of axonal projections. In the present work, we focus on projections reaching the MEC, as this projection represents the most prominent output of PreS neurons (*Caballero-Bleda and Witter, 1993*; *Honda and Ishizuka, 2004*).

A representative experiment is shown in *Figure 4*. Here, the morphology of 2 identified HD cells has been reconstructed (*Figure 4A*); these were pyramidal neurons located in L3 with apical dendrites reaching the pial surface of PreS. These cells sent an axon to the angular bundle; in few instances, the axon split in two branches, one of which travelled caudo-medially (see below) and the other one rostrally (*Figure 4B*, asterisk; see also *Abbasi and Kumar, 2013*) [although the latter branches were not traced further in the present study, we speculate they might target the contralateral MEC, in line with double-retrograde experiments showing partial overlap between contra- and ipsilateral projecting L3 PreS neurons (not shown)]. Caudally-travelling axonal branches often made a sharp turn within the angular bundle before exiting into the deep layers of MEC, where sparse axonal branching could be typically observed. Most axons branched upon entry into L3 and displayed a high density of small axonal varicosities (*Figure 4B,C*). Few branches coursed through L2 and extended within the deep portion of L1, where larger boutons could typically be observed (*Figure 4C*).

In total, 8 long-range axonal projections from identified HD cells could be recovered (median HD index, 0.9; range 0.64–0.96, n = 8; *Figure 4D*). All of them reached the ipsilateral MEC where they generally showed a layer-specific distribution: compared to L2/1 and deep layers, most axonal length was observed within MEC L3 (total axonal length, 2.19 ± 3.38 mm in L3 versus 0.44 ± 0.57 mm in L1, L2 and deep layers; n = 8, p=0.025; *Figure 4D*). The layer-selective branching pattern of the reconstructed single axons is in line with anterograde tracing experiments, which showed that most PreS afferents are observed within MEC L3 (*Caballero-Bleda and Witter, 1993*; *Honda and Ishizuka, 2004*; and data not shown). Altogether, our data provide an anatomical demonstration that the MEC receives HD inputs from upstream PreS L3 neurons, and that HD inputs are arranged according to layer-specific gradients within the MEC.

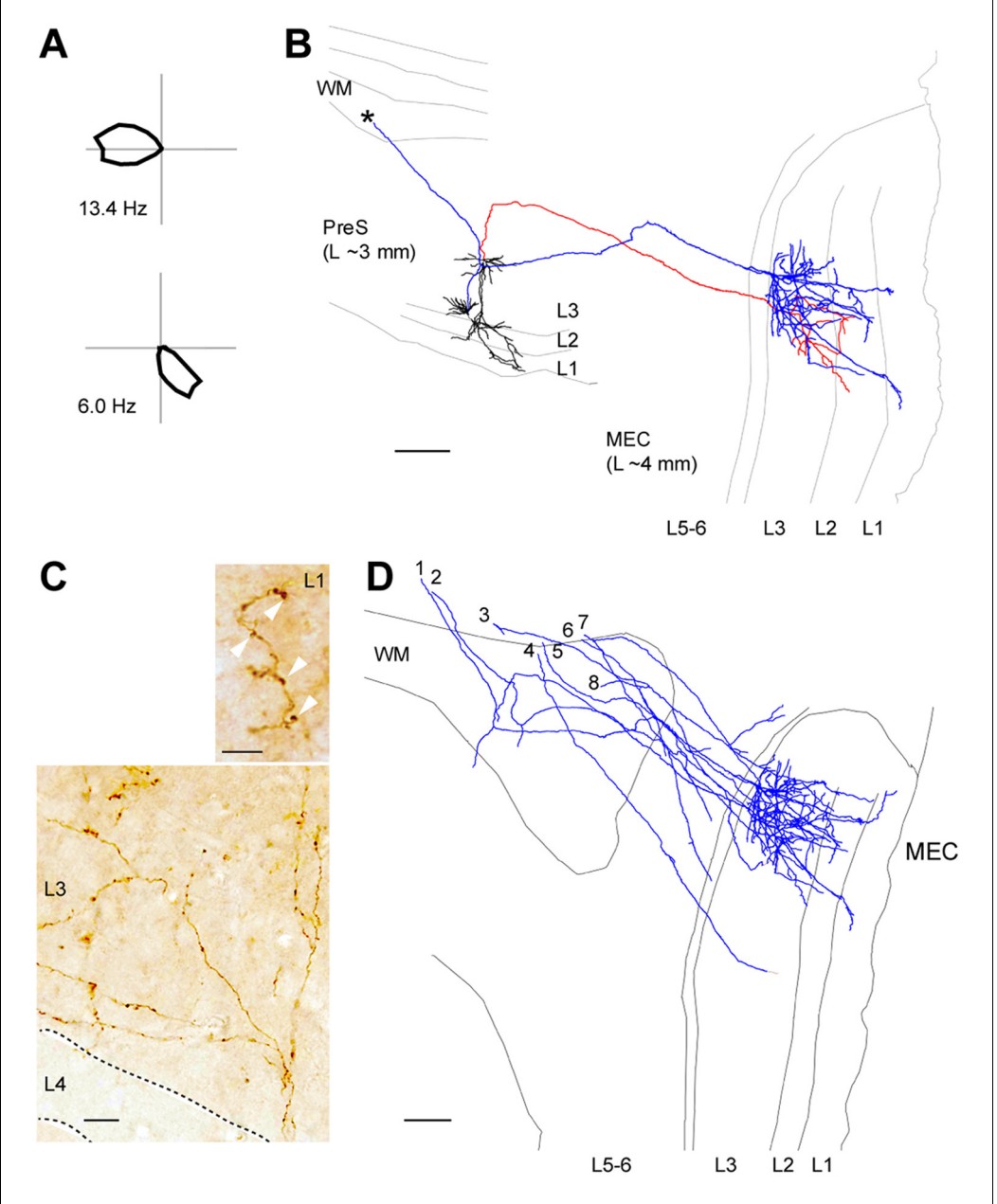

**Figure 4.** Long-range axonal projections of identified PreS HD cells to MEC. (**A**) Polar plots showing firing rate as a function of HD for the two neurons shown in (**B**). (**B**) Morphological reconstruction of two representative layer 3 pyramidal HD cell (dendrites, black; axons, red and blue) which send long-range axonal projections to MEC. Grey lines indicate the outline of the sections relative to the PreS (~3 mm lateral from midline) while axons are aligned to the target area (~4 mm lateral from midline). WM, white matter. Asterisk indicates the rostrally-travelling axonal branch. Scale bar: 200 μm. (**C**) High-magnification micrograph of a DAB stained axon form an identified PreS HD cell, showing branching upon entry in MEC L3. Note the axonal varicosities present in MEC L3 (bottom) and L1 (arrowheads, top). Scale bars, 20 μm (bottom) and 5 μm (top). (**D**) Morphological reconstruction of long-range axonal projections from identified PreS HD cells (n = 8 axons from 8 neurons; blue) which were traced until the superficial layers of MEC. Scale bar: 200 μm.

## HD selectivity and morphological properties of L3 and L2 neurons

The distribution of identified HD cells appeared to follow a layer-specific distribution, since most neurons were recovered in L3 (18/25 identified HD cells) and none in L2. To further investigate this issue, we sought to target juxtacellular recordings to L2. Since this layer is a relatively thin cortical structure - which makes 'blind' juxtacellular targeting particularly challenging- we first explored whether there are electrophysiological signatures of PreS L2, which could enable its selective targeting by juxtacellular procedures. To this end, we employed extracellular recording techniques and monitored multi-unit spiking and local field potential (LFP) activity during electrode penetrations orthogonal to the PreS layers. We found that in awake animals (and to some extent also in anesthetized animals; not shown) L2 could be reliably localized based on two extracellular signatures; first, we often observed an increase in multi-unit spiking activity upon entry into L2, which could possibly result from the relatively higher cellular density within this layer (*Figure 1B*). Second, the transition from L2 to L1 could always be reliably identified, due to the sharp cessation of spiking activity occurring upon entry into L1. Indeed in 4 out of 4 experiments, where electrode locations were confirmed relative to electrolytic lesions (see Materials and methods), we could reliably identify the location of PreS L2 (not shown), indicating that these electrophysiological signatures could be used to successfully target PreS L2.

We thus took advantage of these electrophysiological signatures for targeting juxtacellular recordings to PreS L2. During individual electrode penetrations, multiple consecutive neurons could be typically recorded juxtacellularly across PreS layers. While HD cells were commonly found before the cortical depth of L2 -assessed by prior extracellular mapping- neurons sampled within L2 discharged independently from the direction the rat was facing during passive rotation. These observations were confirmed by juxtacellular labeling, as shown in two representative recordings from identified L2 neurons (*Figure 5*). The basal dendrites of the first neuron, which was calbindin-positive (*Figure 5A*), were largely confined within L2, while the apical dendritic branches covered a large territory within L1. This neuron fired irrespectively of the direction the animal was facing during passive rotation (*Figure 5B,C*; p=0.35). The second neuron also displayed basal dendrites largely confined to L2, a multipolar apical dendritic tree extending into L1, and was calbindin-negative (*Figure 5D*). This neuron was also not tuned to HD (*Figure 5E,F*; p=0.67). In both neurons (*Figure 5A,D*) an axon could be traced within the angular bundle: these axons however followed a different route compared to L3 pyramidal cells (*Figure 4*), as they travelled medially (rather than laterally) within the angular bundle - possibly towards contralateral projection targets, in line with tracing experiments (*Figure 1E–G*).

The physiological differences between PreS L2 and L3 neurons are summarized in *Figure 6*. All identified L2 and L3 neurons included in the analysis displayed 'regular' firing patterns and broad spike waveforms (*Figure 2—figure supplement 1A*); the neurons where morphology could be assessed displayed pyramidal or 'pyramidal-like' morphologies (see Materials and methods; *Figure 6A*) and spiny dendrites (*Figure 6B*) - features classically associated with principal (glutamatergic) cell identity. Both the strength of HD modulation (*Figure 6C*) and the proportion of HD cells were significantly lower in L2 compared to L3 (0/11 in L2 versus 18/25 in L3, p<0.001, Fisher's Exact Test). Notably, while average firing rates did not differ (L2, 2.5 ± 2.5 Hz; L3, 2.4 ± 2.6 Hz; p=0.245), spiking rhythmicity in the theta-frequency range (4–12 Hz) as assessed by a standard 'theta index' (*Yartsev et al., 2011*) was more prominent among L2 than L3 neurons (*Figure 6D,E*; we note that in our head-fixed preparation, theta activity presumably reflects immobility-related type-II theta [*Shin, 2010*, *Tai et al., 2012*] since animals were not actively moving during passive rotation). In line with previous work from freely-moving animals (*Taube et al., 1990a*; *Boccara et al., 2010*; *Tukker et al., 2015*), theta-rhythmicity was very sparse among principal neurons (*Figure 6E*), and the only statistically-significant theta-rhythmic discharges (as assessed by a shuffling procedure; see Materials and methods) were selectively contributed by L2 cells (4 out of 4). Theta-rhythmic spiking patterns were contributed by both calbindin-positive and calbindin-negative neurons (*Figure 6E*), and average theta-indices did not differ significantly between the two cell classes (calbindin-positive, 2.3 ± 2.3, n = 3; calbindin-negative, 3.8 ± 2.5, n = 6; p=0.54; we note however that the small dataset of identified calbindin-positive neurons prevents rigorous assessment of structure-function relationships). The electrophysiological differences between L2 and L3 neurons were not accounted for by biases in rotational parameters, since average angular velocities (L2, 0.96 ± 0.31 rad/s; L3, 0.96 ± 0.33 rad/s; p=0.8), accelerations (L2, 1.41 ±

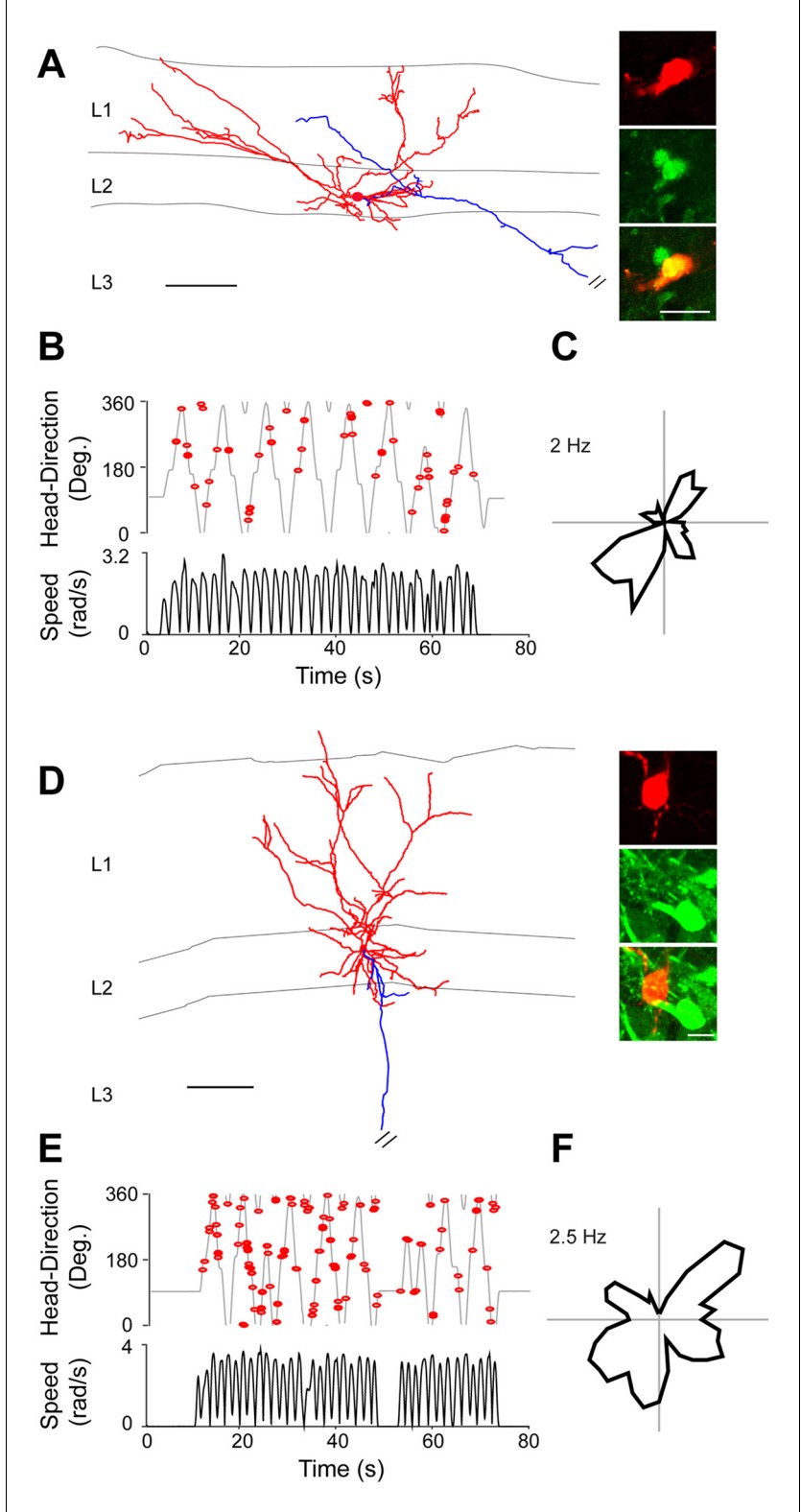

**Figure 5.** Non-directional spiking patterns of identified L2 PreS neurons. (**A**) Left, morphological reconstruction of a representative calbindin-positive layer 2 neuron (dendrites, red; axon, blue) recorded during passive rotation. Scale bar: 100 μm. Right, close-up magnifications of the cell's soma (red, top panel) positive for calbindin immunoreactivity (green, middle panel) and overlay (bottom panel). Scale bar: 20 μm. (**B**) Angular HD (top) and angular speed (bottom) as a function of time. Spikes (red dots) are indicated. (**C**) Polar plots showing firing rate as

*Figure 5 continued on next page*

*Figure 5 continued*

a function of HD for the neuron in (**A**). Peak firing rate is indicated. (**D–F**) same as **A–C** but for a representative -negative L2 neuron. Scale bars in D: 100 μm (left) and 10 μm (right).

0.82 rad/s$^2$; L3, 1.24 ± 0.89 rad/s$^2$; p=0.4) and decelerations (L2, −1.27 ± 0.74 rad/s$^2$; L3, -1.12 ± 0.81 rad/s$^2$; p=0.3) were not significantly different between L2 (n = 11) and L3 (n = 25) recordings. Notably, the spike waveforms of L2 neurons differed significantly from that of L3 cells (*Figure 6F*), as they showed on average a significantly longer duration (as assessed by spike half-width) and more pronounced negativity (*Figure 6G*). Altogether, these data indicate that PreS L2 and L3 neurons can be differentiated according to spike waveform features, HD modulation and temporal spiking properties within the theta-frequency range (see *Figure 6* and *Figure 6—source data 1*).

The present data thus reveal a cell-type and layer specificity of the HD representation within PreS circuits. Together with the different projection targets of L2 and L3 neurons (*Figure 1*), these data point to different routing of directional and non-directional information from the superficial PreS layers to downstream cortical areas (see *Figure 6—figure supplement 1*).

## Discussion

The PreS is widely recognized as a key structure in the cortical representation of HD. Here we show that the superficial layers of the rat PreS are composed of molecularly- and morphologically-distinct principal cell populations, which can be differentiated according to long-range projection targets. Temporal and directional firing properties are differentially distributed among these neurons, with L3 pyramidal cells being predominantly modulated by HD, and L2 neurons' spiking being largely unaffected by HD but significantly entrained by theta oscillations. These findings closely resemble the cytoarchitectonic and functional architecture of the MEC; specifically, also in MEC (i) calbindin-positive neurons are clustered and project (at least to some extent) to the contralateral homologue area (*Varga et al., 2010*; *Fuchs et al., 2016*), and (ii) L2 and L3 neurons differ in morphological, electrophysiological, functional properties and projection targets (*Kitamura et al., 2014*; *Ray et al., 2014*). Notably, both Pres and MEC have been shown to contain the same types of spatially-modulated neurons, albeit in different proportions (*Boccara et al., 2010*). The common basic architecture of PreS and MEC circuits could point to similar mechanisms supporting the generation of spatial firing within these areas, as proposed by previous authors (*Boccara et al., 2010*).

Our experimental design, based on passive rotation of head-fixed rats (see Materials and methods) was optimized for selectively targeting HD cells. Our experiments confirmed earlier work which indicated that HD responses are largely preserved under these conditions (*Zugaro et al., 2001*; *2002*; *Shinder and Taube, 2011*; *2014*). Indeed, the abundance and general properties of PreS HD neurons were very similar to the ones reported from freely-moving animals, thus pointing to a largely intact HD system which is uncoupled from voluntary animal locomotion (see also *Winter et al., 2015b*). Our approach thus enabled efficient identification and labeling of HD neurons (*Figure 3*) but prevented assessment of spatial firing properties. Hence, the spatial firing patterns of the 'non-directional' L2 neurons remain to be established. We note that the lack of directionality in PreS L2 might be due to the lack of HD inputs from the dorsal thalamus, in line with tracing experiments indicating that thalamic inputs largely avoid PreS L2 (*van Groen and Wyss, 1990b*; *1995*; *Shibata, 1993*). An intriguing possibility is that L2 neurons could contribute the spatial signals which have been previously recorded among PreS units (i.e. grid and border cells; *Boccara et al., 2010*; *Winter et al., 2015b*). Based on the known correlation between spiking theta-rhythmicity and grid activity (*Boccara et al., 2010*; *Brandon et al., 2011*; *Koenig et al., 2011*) and the fact that under our recording configuration, theta-rhythmic responses were almost exclusively contributed by L2 PreS neurons (*Figure 6D,E*), we speculate that L2 could be the principal source of grid activity in PreS - as it is the case in MEC (*Boccara et al., 2010*). Future approaches involving either juxtacellular labeling (*Tang et al., 2014*) or genetic targeting in freely-moving animals will be required for testing this hypothesis.

Reconstructions of long-range axonal projections from functionally-identified PreS HD cells provided direct anatomical evidence that the MEC receives HD inputs, complementing earlier evidence

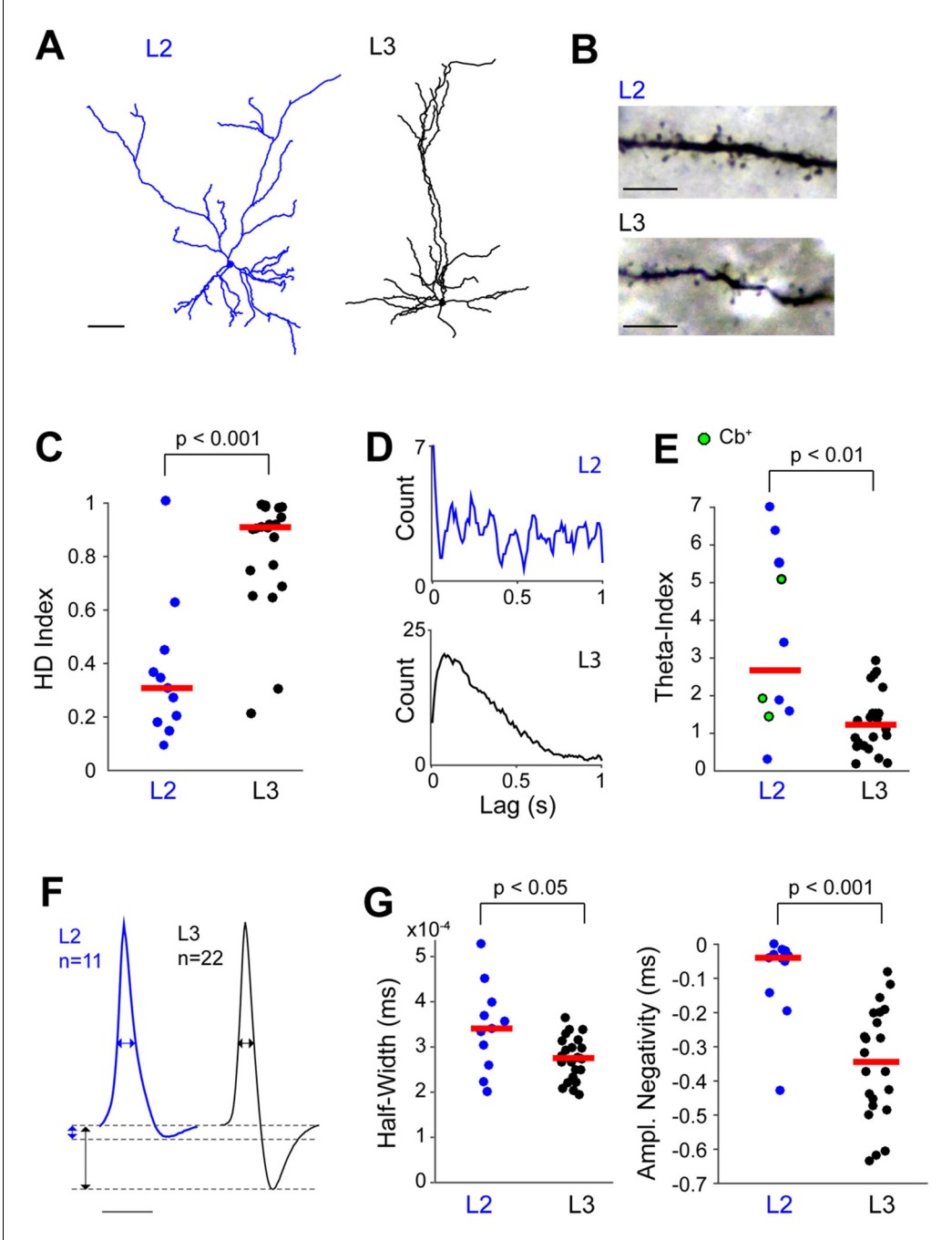

**Figure 6.** Morphological and electrophysiological properties of L2 and L3 PreS neurons. (A) Morphological reconstruction of a representative L2 (blue, left) and L3 (black, right) neuron, recorded during passive rotation. Scale bar = 50 µm. (B) Representative high-magnification pictures of a dendritic segment of a L2 (top) and L3 (bottom) neuron. Note the presence of spines in high density in both dendrites. Scale bars = 10 µm. (C) HD indices for all identified L2 (n = 11) and L3 neurons (n = 22). Three L3 neurons were silent, and hence not included in the analysis. Horizontal red lines represent medians and the p value is indicated (Mann-Whitney U test). (D) Representative spike-autocorrelogram for an identified L2 (top) and L3 neuron (bottom). Note the theta-rhythmicity of spiking for the L2 neuron. (E) Theta indices for all identified L2 (n = 10) and L3 neurons (n = 22) which met inclusion criteria for the theta analysis (see Materials and methods). Horizontal red lines represent medians and the p value is indicated (Mann-Whitney U test). (F) Average spike waveforms for L2 (blue, n = 11) and L3 (black, n = 22) neurons. Three L3 neurons were not included in the analysis since they were silent. Horizontal and vertical double-arrowheads indicate spike half-widths and spike negativity amplitudes, respectively. Scale bar

*Figure 6 continued on next page*

*Figure 6 continued*

= 1 ms. (**G**) Spike half-widths (left) and spike negativity amplitudes (right) for L2 (n = 11) and L3 (n = 22) neurons. Horizontal red lines represent medians and the p value is indicated (Mann-Whitney U test).

The following source data and figure supplement are available for figure 6:

**Source data 1.** Electrophysiological properties of identified L2 and L3 PreS neurons.

**Figure supplement 1.** Schematic representation of structure-function relationships within the superficial layers of PreS.

(*Tukker et al., 2015*) and in line with predictions from computational models (*Burak and Fiete, 2006*; *McNaughton et al., 2006*; *Bush and Burgess, 2014*). HD inputs are thought to be critically involved in the generation of grid activity; the long-range HD circuits we describe in the present study (*Figure 4*) could provide –together with the parasubiculum (*Tang et al., 2016*) - one source for HD inputs into the grid system. The distribution of axonal length and axonal varicosities (*Figure 4C,D*) indicated that most synaptic contacts are likely to occur within MEC L3. Although axonal-bouton distribution is typically in large agreement with connectivity inferred by direct methods, it remains to be established whether MEC L3 neurons are indeed the prime recipient of HD inputs (see *Canto et al., 2012*). An intriguing observation is that the representation of HD appears to be much sparser in MEC L3 (*Giocomo et al., 2014*; *Tang et al., 2015*) compared to its presynaptic inputs structure (i.e. PreS L3), arguing against a simple feed-forward inheritance of HD coding. It will be crucial to resolve the signal transformation occurring in MEC L3 (*Tang et al., 2015*) for understanding the layer specific contribution of MEC circuits to spatial coding.

The cellular and circuit organization of PreS is likely optimized for subserving a specific function during navigation and episodic memory. Inactivation and lesion studies have indicated that the PreS might be critically involved in the stability of spatial representations (*Taube et al., 1992*; *Goodridge and Taube, 1997*; *Calton et al., 2003*; *Taube, 2007*) by binding visual landmark information to the HD representation (*Vogt and Miller, 1983*; *McNaughton et al., 1991*; *Goodridge and Taube, 1995*; *Knierim et al., 1995*; *Taube, 2007*). In this context, the connection from PreS L2 to RS cortex (*Figure 1G*) – an area known to receive strong direct inputs from primary visual cortex (*Ding, 2013*) – could point to L2 as the site where visual information is processed and integrated into the PreS HD map. Future work will be required for dissecting the contribution of PreS and RS neurons to this computation, which is crucial for the stable expression of cognitive representations of space.

## Materials and methods

### Histological analysis, histochemistry and immunohistochemistry

At the end of each recording, the animal was euthanized with an overdose of pentobarbital and quickly perfused transcardially with 0.1 M phosphate-buffered saline followed by a 4% paraformaldehyde solution. Brains were removed from the skull, immersed in fixative for at least one day and cut with vibratome or cryostat (prior cryo-protection step in 30% sucrose) to obtain 50–70 µm thick parasagittal sections. To reveal the morphology of juxtacellularly labeled cells (i.e. filled with neurobiotin or biocytin, see below), brain slices were processed with streptavidin-546 or 488 (Life Technologies, UK). Immunohistochemical stainings for Calbindin (Monoclonal or Rabbit anti Calbindin D28-k, 1:2000; Swant, Switzerland), Paravalbumin (Monoclonal anti paravalbumin, 1:3000; Swant), Wolframin (Rabbit anti Wfs-1, 1:500; ProteinTech, UK) and NeuN (Anti-NeuN A60, 1:1000; Millipore, USA) were performed on free-floating sections. Immunohistochemical images were acquired by epifluorescence (Axio imager Zeiss) or confocal (Zeiss LSM 710) microscopy, and the analysis was performed with Neurolucida software (MBF bioscience). After fluorescence images were acquired, the neurobiotin/biocytin staining was converted into a dark DAB reaction product. Some sections underwent $Ni^{2+}$-DAB enhancement protocol (*Klausberger et al., 2003*). Zinc staining was essentially performed as previously described (*Danscher, 1981*; *Ichinohe and Rockland, 2004*). Briefly, after perfusion

with a solution containing sodium sulfide, brain sections were washed thoroughly with 0.1 M and 0.01 M phosphate buffer solutions. Sections were then developed by exposing them to a solution containing gum arabic, citrate buffer, hydroquinone and silver lactate for 60–120 min in the dark at room temperature. Development of reaction products was terminated by rinsing the sections in 0.01 M phosphate buffer and subsequently several times in 0.1 M phosphate buffer.

## Retrograde neuronal labeling

Retrograde tracer solutions containing Cholera Toxin Subunit B- Alexa Fluor 488 or 546 conjugates (Life Technologies) (CTB; 0.8% w/vol in PB 0.1 M) were injected in 200-–250 g rats under ketamine/ xylazine anesthesia. Briefly, animals were placed in a stereotaxic apparatus, and a small craniotomy (<1 mm$^2$) was performed at the coordinates for targeting the MEC (see *Burgalossi et al., 2011*; *Ray et al., 2014*), dorsal PreS (lambda coordinates: 0.0 mm AP, 3.3 mm ML, −3.0 mm DV) or RS29 (lambda coordinates: −1.0 mm AP, 3.3 mm ML, −3.0 mm DV). Injections in RS29 (n = 4) were centered on, but not restricted to, the caudal portion of the RS29, bordering rostrally with the PreS. The border between PreS and RS29 were confirmed by calbindin staining (see *Figure 1—figure supplement 2*). We note that this region has been previously referred to as 'area retrosplenialis 29e' (*Blackstad, 1956*; *Haug, 1976*; *Slomianka and Geneser, 1991*) or 'triangular region' (*Ding, 2013*). PreS injections (n = 3) were centered on L2; to this end, prior to injection, PreS layer 2 was localized by electrophysiological mapping with low-resistance electrodes (1–3 MΩ), based on characteristic signatures of the multiunit spiking activity (see Results). Glass electrodes with a tip diameter of ~20–40 μm filled with CTB solution were then lowered into the target region. To avoid diffusion of the tracer during electrode penetration, the tip of the pipette was front-filled with a small amount of Ringer solution. Typically, small amounts of tracer solutions (~0.3–0.8 μl) were then slowly injected using positive pressure. After the injections, the pipettes were left in place for several minutes and slowly retracted. The craniotomy was closed by application of silicone (Kwik-cast, World Precision Instruments) and animals survived for 3–5 days before being euthanized and transcardially perfused. Both hemispheres were cut into 60–70 μm thick parasagittal slices and analyzed with epifluorescence and/or confocal microscopy. When necessary, immunochemical staining for calbindin was performed to outline the cytoarchitecture of the superficial PreS layers.

## Analysis of anatomy data

Retrogradely-labeled PreS neurons (*Figure 1C–G* and *Figure 1—figure supplement 3*) and NeuN/ calbindin-positive PreS L2 neurons (*Figure 1B*) were manually counted on z-stacks with the Neurolucida software. For tracing experiments, neurons were counted from six parasagittal sections encompassing the medio-lateral extent of the dorsal PreS. Neuronal reconstructions of juxtacellular labeled cells were performed manually with the Neurolucida software and displayed as 2-dimensional projections. The projection planes for the cells in *Figure 5* were optimized (by rotation along the dorsoventral plane) in order to obtain optimal display of apical dendritic branches. For displaying long-range axonal projections of PreS HD cells (*Figure 4*), PreS cells were registered relative to the parasagittal PreS section containing their somato-dendritic compartment, while axons were superimposed on the reconstruction of more lateral parasagittal sections containing MEC at a typical mediolateral level (*Figure 4D*).

## Juxtacellular recordings

Experimental procedures for obtaining juxtacellular recordings, signal acquisition and processing and animal tracking in awake, head-fixed animals were essentially performed as recently described (*Diamantaki et al., 2016*; *Houweling et al., 2010*). Briefly, recordings were made from male Wistar rats (~150–250 g). Glass pipettes with resistance 4–6 MΩ were filled with extracellular (Ringer) solution containing in mM: 135 NaCl, 5.4 KCl, 5 HEPES, 1.8 CaCl$_2$ and 1 MgCl$_2$ (pH is adjusted to 7.2) plus Neurobiotin (1.5–3%; Vector Laboratories, UK) or Biocytin (1.5–3%; Sigma-Aldrich, Germany). Osmolarity was adjusted to 290–320 mOsm.

We used head-restrain and passive-rotation procedures following the work of *Shinder and Taube (2011)*; (*2014*), i.e. animals were head-restrained onto a rotatable platform, which was rotated manually by the experimenter. For these experiments, animals were pre-implanted with a metal post and a recording chamber under ketamine/xylazine anesthesia. After a recovery period (~3–4 days)

animals were slowly habituated to head-fixation and to the rotation of the apparatus. Habituation and recordings were performed under slightly-dimmed ambient illumination in the 'cue-rich' environment of the laboratory setting. Thus, both during habituation and recordings, the rats had visual access to proximal cues available in the immediate vicinity (e.g. computer screens, cold-light source, stereomicroscope) and distal cues (i.e. Faraday cage, ceiling, curtains), including the experimenter, which was always located in the same relative position during the passive rotation experiment. These cues were thus the most likely source of 'anchoring' stability to HD firing (see e.g. *Knierim et al., 1995* for review). The stability of HD responses in the dark (i.e. in the absence of visual cues) has not been tested in the present study. Craniotomies ($<1$ mm$^2$) were performed at the coordinates for targeting the dorsal PreS (0–0.5 mm posterior and 3–3.7 mm lateral from Lambda). Before juxtacellular recordings, mapping experiments with low-resistance electrodes (0.5–1 M$\Omega$) were performed to precisely estimate the location of the PreS, and of PreS L2. In a subset of preliminary experiments, the location of L2 was confirmed by aligning Tungsten electrode tracks to anatomically-verified electrolytic lesions (n = 4), essentially as previously described (*Beed et al., 2013*).

Juxtacellular labeling was performed by using standard labeling protocols (*Pinault, 1994*; *1996*) and modified procedures, which consisted in rapidly breaking the dielectric membrane resistance by short (1–2 ms) 'buzz-like' current pulses, which provided rapid access to cell entrainment by juxtacellular current injection (i.e. 200 ms square current pulses, *Pinault, 1994*; *1996*). After cell labeling, animals were either immediately perfused for anatomical analysis, or returned to their home cage and perfused ~4–12 hr following labeling. In order to maximize axonal recovery, in some experiments multiple neurons were labeled; the sparse labeling typically allowed unequivocal assignment of the identified cells, based on positional coordinates and recording depth. In total, 54 neurons (48 principal cells and 6 interneurons) were labeled and recovered in PreS. In 36 out of 48 cases, the morphology of principal neurons could be assessed; in the remaining cases, morphology could not be assessed as only the soma and/or proximal dendrites were recovered. Cells were classified as 'pyramidal' if a pyramidal-shaped soma and at least a prominent apical dendrite could be identified. Non-pyramidal, 'multipolar' morphologies were classified based on the proximal dendritic arrangement and the lack of prominent apical dendrite(s). Within L2, principal cells generally displayed 'pyramidal-like' morphologies, with often multiple apical dendrites branching extensively within L1. Ten out of 11 L2 neurons were tested for calbindin expression (3 calbindin-positive and 7 calbindin-negative neurons). Identified neurons were classified as interneurons (n = 6) based on classical morphological features (e.g. thin, smooth and often 'beaded' dendrites; see *Ascoli et al. (2008)* for review). Two FS interneurons were tested for PV expression, and were positive (one neuron shown in *Figure 2—figure supplement 1E*). In 8 neurons, a long-range axon was traced till the ipsilateral MEC. The quality in axonal filling differed among the individual cases, and in general it cannot be assured whether even in the best-filled examples, thin axonal branches were missed due to incomplete filling. Nevertheless, the presence of axonal boutons (which were always associated with terminal axonal branching) within MEC (*Figure 4*) provides anatomical demonstration that HD inputs target MEC neurons with a bias for MEC L3 (*Figure 4*).

The juxtacellular voltage signal was acquired via an ELC-03XS amplifier (NPI Electronic), sampled at 20 kHz by a LIH 1600 data-acquisition interface (HEKA Electronic) under the control of PatchMaster 2.20 software (HEKA Electronic) or Spike2 v8.02 software and Power1401-3 data-acquisition interface (CED, UK). Extracellular signals were acquired via an EXT-HS-M amplifier (NPI Electronic); either broad-band (e.g. for LFP) or band-pass (i.e. for spikes) signals were acquired by filtering the extracellular signals via a DPA-2F2 filter unit (NPI Electronic). The orientation of the rat's head was tracked using a LED placed on the back of the turntable, in line with the sagittal plane of the animal. Animal tracking was performed by acquiring a video (25 Hz frame rate) with the IC Capture Software (The Imaging Source).

## Analysis of electrophysiology data

Spike signals from juxtacellular traces and a few extracellular units (n = 6) were isolated by using principal component analysis, essentially as previously described (*Burgalossi et al., 2011*). The bursting index (see *Figure 6—source data 1*) was defined as the sum of spikes with an ISI < 6 ms, divided by the number of spikes. A single white LED, positioned on the rotating apparatus, was used for extracting the HD angle and the angular velocity. The angular velocity was calculated based on smoothed X and Y coordinates of the tracking (averaged across a 600 ms rectangular sliding window). A linear

velocity cutoff (1 cm/s) was applied for isolating periods of rest from rotational movement, and only spikes during movement were included in the theta, speed and HD analysis (see below).

The theta-index was computed as in *Yartsev et al. (2011)*. Briefly, theta-rhythmicity of spiking was determined by first calculating the spike train's autocorrelation for each cell using a 10 ms bin size. The power spectrum obtained by calculating the Fourier Transformation on the autocorrelation was used to measure the modulation strength in the theta band (4–12 Hz). The theta index was defined as the average power within 1 Hz of the maximum of the autocorrelation function in the theta band divided by the average power between 1 and 50 Hz. Only recordings with >20 spikes were included in the theta-rhythmicity analysis (n = 10 L2 and n = 22 L3 neurons; *Figure 6D,E*). Statistical significance of theta-rhythmicity was evaluated with a shuffling test (essentially as described by *Yartsev et al. (2011)*), which was performed on a cell-by-cell basis; for each trial of the shuffling procedure, individual spike times were randomly time-shifted. For each permutation, the theta-index was calculated and the procedure reiterated 1000 times. The significance value for each cell was assessed based on the resulting null distribution, i.e. a neuron was defined as significantly theta-rhythmic if the theta-index was >95th percentile of its corresponding null distribution.

Speed analysis was performed as in *Kropff et al. (2015)*. Briefly, a speed score was defined as the Pearson's product-moment correlation between the instantaneous firing rate and the rat's instantaneous angular velocity. A neuron was defined as significantly modulated by angular velocity if its speed score was >95th of the null distribution, generated by a shuffling procedure (1000 permutations per cell) essentially as in *Kropff et al. (2015)*. The firing rate and angular velocities were calculated with 40 ms bins, coinciding with the frames of the tracking camera. Angular acceleration and deceleration were calculated as $\alpha = d\omega/dt$, where $\omega$ is the angular velocity and $dt$ the time between two frames (40 ms). For calculating the number of inversions during passive rotation, an inversion was defined as a sign change of the difference between two consecutive angles, if larger than $\pi$ radians.

In total, we recorded n = 310 PreS neurons in awake, head-fixed rats during passive rotation, where all HD bins (n = 36) were visited at least once (as in *Tukker et al., 2015*). Recordings (or portions of recordings) were cellular damage was observed in the electrophysiology were excluded from the analysis (as in *Pinault, 1996*; *Herfst et al., 2012*). To quantify HD tuning, we divided the number of spikes by the occupancy for each HD bin. The HD index of a cell was defined as the average Rayleigh vector over all bins, essentially as previously described (*Boccara et al., 2010*; *Tukker et al., 2015*). Significance was evaluated with a shuffling test, which was performed on a cell-by-cell basis; for each trial of the shuffling procedure, the entire sequence of spikes was randomly time-shifted. For each permutation, the HD Index was calculated and the procedure reiterated 1000 times. The significance value for each cell was assessed based on the resulting null distribution, i.e. a neuron was defined as HD cell if the HD Index was > 95th percentile of its corresponding null distribution. For recordings in which each HD bin was sampled in each half of the recording (n = 181 out of 186 HD cells), we quantified the stability of the HD tuning by generating separate tuning curves for the first and second half of the recording time and calculating Pearson's linear correlation coefficient.

For all experiments, sample sizes were estimated based on previously published data using similar procedures (*Ray et al., 2014*; *Tang et al., 2014*; *2015*). Statistical significance was assessed by a two-sided Mann-Whitney nonparametric test with 95% confidence intervals.

## Acknowledgements

We thank Alexandra Eritja for excellent assistance with anatomy experiments, Fereshteh Zarebidaki for contributing to anatomy experiments and data analysis, Thomas Klausberger and Erzsebet Borok for helpful advices on DAB-enhancement procedures, Maria Diamantaki, John Tukker and Robert Naumann for helpful comments on earlier versions of the manuscript.

# Additional information

## Funding

| Funder | Author |
| --- | --- |
| Deutsche Forschungsgemeinschaft | Patricia Preston-Ferrer<br>Stefano Coletta<br>Markus Frey<br>Andrea Burgalossi |

The funders had no role in study design, data collection and interpretation, or the decision to submit the work for publication.

## Author contributions

PP-F, SC, Acquisition of data, Analysis and interpretation of data, Drafting or revising the article; MF, Analysis and interpretation of data, Drafting or revising the article; AB, Conception and design, Acquisition of data, Analysis and interpretation of data, Drafting or revising the article

## Author ORCIDs

Andrea Burgalossi, http://orcid.org/0000-0003-0039-3599

## Ethics

Animal experimentation: All experimental procedures were performed according to the German guidelines on animal welfare and approved by the local institution in charge of experiments using animals (Regierungspraesidium Tuebingen, permit numbers CIN2/14, CIN5/14 and CIN8/14).

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
