## [Decision Letter]

Thank you for submitting your work entitled "Anatomical Organization of Presubicular Head-Direction Circuits" for consideration by *eLife*. Your article has been reviewed by three peer reviewers, and the evaluation has been overseen by a Howard Eichenbaum as the Reviewing Editor and Timothy Behrens as the Senior Editor. The following individuals involved in review of your submission have agreed to reveal their identity: Kate Jefferey (peer reviewer).

The reviewers have discussed the reviews with one another and the Reviewing Editor has drafted a list of required and recommended changes and experiments that we feel must be considered if this work is to be published in *eLife*.

Head-direction (HD) cells form the building block of the navigation system. Unveiling their dynamics and circuitry is vital to further our understanding of how the cognitive map forms within the medial entorhinal cortex (MEC) and the hippocampus. In the present manuscript, Preston-Ferrer and colleagues present novel findings on the circuit underlying the processing of the HD information in the pre-subiculum (PreS), an important relay of the HD signal to the MEC. Using juxtacellular recordings from awake, head-restrained mice, the authors present data from a large amount of neurons and were able to reconstruct for identified HD cells in vivo their morphology, anatomy and axonal projections. The paper presents three levels of anatomical differentiation within the PreS: i) HD cells were identified within Layer 3, not Layer 2, ii) Layer 3 cells (including HD cells) project to the ipsilateral MEC, Layer 2 neurons project to the contralateral PreS; iii) Layer 2 showed more theta modulation than Layer 3 cells. These results are interesting but the data, in their present form, do not fully support the novel findings.

Major issues:

1) The authors have showed that Layer 3 cells project to the MEC, using both histological reconstruction and retrograde labeling. Based on the reconstruction of 8 identified HD cells, they show quite clearly that they target mainly the superficial layers of the MEC. This is one of the most important findings of the manuscript. Indeed, this is surprising and not really expected, as the authors admitted themselves, as (i) HD cells are more prominent in the deep layers of the MEC than in the superficial layers (where they are virtually absent) and (ii) a previous report (Tukker et al., 2015; but only one reconstructed axon was shown there) reported a projection from PreS layer 3 HD cell to the deep layers. Why haven't the author tried an anterograde labeling from the PreS to demonstrate definitively their claim? Do only HD cells from layer 3 project to the MEC or all cells from layer 3? (i.e. did the authors have reconstructed the axons of non-HD cells from PreS layer 3)? Did these 8 HD cells have strong HD index?

2) The authors did not adequately describe the apparatus used for the head-fixation. Were the animals free to run on a treadmill or on a wheel? If so, did the author had access to the speed of the animal? This is a clear limitation of the head-fixed experiment compared to freely moving juxtacellular recording (again, Tukker et al., 2015). The fact that Layer 2 neurons are theta modulated suggests that they may be modulated by speed, and that will be the demonstration of a nice functional and anatomical segregation between the HD and speed signals. Perhaps the authors should first try to correlate the firing rate with angular speed?

3) The higher theta rhythmicity in layer 2 was mostly explained by a few strongly theta modulated cells (Figure 7). Is there any evidence that these cells were different from the others? Were they all excitatory pyramidal or stellate cells or is there a chance that some were interneurons? By the way, have the authors recorded from any interneurons in the course of this study?

4) Following up on the previous point, could the authors report more detailed descriptions of the cells they recorded within the two layers: average firing rate (partially reported in the text), peak firing rates for HD cells, waveform width, etc.

5) Is there a relationship between calbindin expression and theta rhythmicity? As the authors reported that these two classes of cells have different output, it will be interesting to show that they are also functionally segregated.

6) The experiments described were performed in head-fixed rats with passive rotation. Such an experimental design has been used before for thalamic recordings, which showed that head-direction cells can be observed under such conditions, but some differences seem to exist to free movement. Because the entire manuscript is based on this preparation, a control experiment for presubicular recordings in the set-up used here should be performed and some neurons should be recorded consecutively during passive rotation and free movement. Of course, this does not require the difficult juxta recording and labelling of the neurons, but could be done with tetrodes or silicon probes, which give more stable recordings and many cells can be recorded simultaneously. This experiment is well within the expertise of the authors and could be done relatively quickly. Importantly, it would give a quantitative measure how the same neurons fire according to head-direction in passive and free movement. This would answer if the same neurons are head-direction cells and how their head-direction tuning changes under both conditions.

7) In the Introduction the authors lay out their aim with: "Specifically, it is currently unknown how HDcells are anatomically organized within the PreS, and whether they project to MEC, as long speculated by computational models (see Giocomo et al., 2014 for review)." However, in a previous paper – the corresponding author of present manuscript was a co-auhor in this older paper – they say "[…]our data do indicate that at least a subset of the MEC-projecting pyramidal cells in layer 3 of the presubiculum is HD" (Tukker et al., J Neurosci 2015). A fair presentation of what has been achieved previously is necessary and should also be reflected in the significance statement.

8) For the presented data, it is key to define the exact border between layer 2 and layer 3, which is not trivial. This should be shown with clear examples and a better description on how exactly it was decided for neurons that were located close to the border.

9) How was the rotation speed compared between different experiments?

10) For the physiological identification of layer, a histological example of a lesion should be shown. Is it really possible to make such a small lesion to determine the border?

11) It would be useful to get a slightly better picture of where exactly in presubiculum the study was done, perhaps with a zoomed-out anatomical illustration, and location on an atlas, so that the exact region is more obvious for anyone who might want to pursue this further. The analysis is focused on MEC but the interconnections with RSC are also very important to understand, as the authors note in the discussion, and the impact statement could be adapted to include this. We would like to know more about which part of RSC (e.g. AP location, layer etc) was targeted and exactly what the labelling patterns were. We also need a lot more details about the behavioural manipulation – how was the animal rotated, with what angular acceleration and velocity, how many reversals, what visual cues were available, etc.

[Editors' note: further revisions were requested prior to acceptance, as described below.]

Thank you for submitting your article "Anatomical Organization of Presubicular Head-Direction Circuits" for consideration by *eLife*. Your article has been reviewed by three peer reviewers, and the evaluation has been overseen by a Reviewing Editor and Timothy Behrens as the Senior Editor. The following individuals involved in review of your submission have agreed to reveal their identity: Adrien Peyrache (Reviewer #1); Thomas Klausberger (Reviewer #2); Kate J Jeffery (Reviewer #3).

The reviewers have discussed the reviews with one another and the Reviewing Editor has drafted this decision to help you prepare a revised submission.

Summary:

The authors have taken great care in adequately addressing the concerns and questions of the referees. The additional experiments have significantly strengthened the manuscript.

Essential revisions:

1) The additional data and figures are particularly useful. Please include one of the sub-parts of Figure 1—figure supplement 1 (micrograph plus line drawing) in the main paper, to help the reader visualize the anatomy.

2) Please complete the presentation with a wiring diagram including the cell types, their properties, distribution and connections, summarizing the take-home message visually (akin to a graphical abstract).

[Editors' note: further revisions were requested prior to acceptance, as described below.]

Thank you for resubmitting your work entitled "Anatomical Organization of Presubicular Head-Direction Circuits" for further consideration at *eLife*. Your revised article has been favorably evaluated by Timothy Behrens (Senior editor), and a Reviewing editor in consultation with the reviewers.

The manuscript has been improved but there are some remaining issues that need to be addressed before acceptance, as outlined below:

The reviewers found the article adequately revised but one reviewer had the following recommendations. The reviewer found the additional data and figures particularly useful and recommended that one of the sub-parts of Figure 1—figure supplement 1 (micrograph plus line drawing) be moved to the main paper to help the readers visualize the anatomy. Also this reviewer suggested that the authors add a wiring diagram including the cell types, their properties, distribution and connections, summarizing the take-home message visually (akin to a graphical abstract).

---

## [Author Response]

*Major issues:*

*1) The authors have showed that Layer 3 cells project to the MEC, using both histological reconstruction and retrograde labeling. Based on the reconstruction of 8 identified HD cells, they show quite clearly that they target mainly the superficial layers of the MEC. This is one of the most important findings of the manuscript. Indeed, this is surprising and not really expected, as the authors admitted themselves, as (i) HD cells are more prominent in the deep layers of the MEC than in the superficial layers (where they are virtually absent) and (ii) a previous report (Tukker et al., 2015; but only one reconstructed axon was shown there) reported a projection from PreS layer 3 HD cell to the deep layers. Why haven't the author tried an anterograde labeling from the PreS to demonstrate definitively their claim?*

First point; we agree with the Reviewers that the present findings are somewhat ‘surprising’, in light of the sparseness of strong head-directionality among MEC L3 neurons (Giocomo et al., 2012; Tang et al., 2015; but see Sargolini et al., 2006; Boccara et al., 2010). Our findings are however not incompatible with existing data since (i) as we pointed out in the discussion, our data do not exclude synapses being made on the apical dendrites of deep-layer neurons (see for example Canto and Witter, J Neurosc 2012) and (ii) grid cells from MEC L2/3 have indeed been shown to receive HD inputs (Bonnevie et al., 2013). These points are referred to in the discussion of the revised manuscript. Second point; we would like to specify that in our previous study (Tukker et al., 2015) a single axon ‘could be traced till the deep layers of MEC’, but did not terminate there (i.e. no terminal branching or axonal boutons were described). Hence, it is likely the single axon of Tukker et al. was en route towards L3 – as the present work (Figure 4), our new tracing experiments (see Figure 7) and previous work indicates (Caballero-Bleda and Witter, 1993, Honda and Ishizuka, 2004). Third point; we have performed the anterograde tracing experiments suggested by the Reviewers (see Figure 7). In line with our single axonal reconstructions, the large majority of axons and boutons were indeed observed within MEC L3 (Figure 7).

Author response image 1.Laminar organization of projections from the PreS to MEC.(**A**) Parasagittal sectionthrough MEC showing anterogradely-labeled axons following injectionof the anterogradetracer BDA -10k (red) in the ipsilateral PreS. Entorhinal layers are outlined via calbindin staining (Cb, green). Note the massive arborization of PreS afferents within L3 of the MEC. (**B** and **C**) show high-magnifications viewof the section shown in **A**.**DOI:**
http://dx.doi.org/10.7554/eLife.14592.015

Due to an already large supplementary material, we have not included these data in the revised manuscript. We however refer to this point in the Results section and cite the relevant literature (Caballero-Bleda and Witter, 1993,Honda and Ishizuka, 2004). If the Editor/Reviewers feel these data should be included in the revised manuscript, we will be happy to do so.

We have addressed this comment by specifying in the revised manuscript that ‘it remains to be established whether MEC L3 neurons are indeed the prime recipient of H D inputs (see Canto and Witter, 2012) (Discussion). In the results we state that ‘the layer-selective branching pattern of the reconstructed single axons is in line with anterograde tracing experiments, which showed that most Pre S afferents are observed within MEC L3 (Caballero Bleda and Witter, 1993, Honda and Ishizuka, 2004 an d data not shown)’

*Do only HD cells from layer 3 project to the MEC or all cells from layer 3? (i.e. did the authors have reconstructed the axons of non-HD cells from PreS layer 3)? Did these 8 HD cells have strong HD index?*

All axons which could be traced till MEC (n=8) carried a significant amount of HD (p <0.05; median H D index, 0.9; range 0.64 – 0.96). We note however that given the small dataset of identified long-range axons, it cannot be concluded that HD signals re the sole input reaching the downstream MEC.

We specify in the revised manuscript that ‘[…]8 long-range axon al projections from identified HD cells could be recovered (median HD index, 0.9; range 0.6 4 – 0.96, n= 8; Figure 4)’ (Results).

*2) The authors did not adequately describe the apparatus used for the head-fixation. Were the animals free to run on a treadmill or on a wheel? If so, did the author had access to the speed of the animal? This is a clear limitation of the head-fixed experiment compared to freely moving juxtacellular recording (again, Tukker et al., 2015). The fact that Layer 2 neurons are theta modulated suggests that they may be modulated by speed, and that will be the demonstration of a nice functional and anatomical segregation between the HD and speed signals. Perhaps the authors should first try to correlate the firing rate with angular speed?*

First, we apologize for not having provided sufficient details about the passive-rotation procedures (see also response to comment 11 below). We have now added more information in the revised Methods, and provide a video of a representative HD cell recorded during passive rotation (Video 1).

Second, we specify that our rats were not on a treadmill, and hence we cannot correlate spiking activity to the animal translational movement. For clarity, we now state in the revised manuscript that under our preparation, theta-activity most likely reflects immobility-related type-II theta, as previous work on passively-rotated rats has indicated (Shin et al., 2010). Following the Reviewers’ suggestion, we have correlated neuronal activity to angular velocity. We followed the criteria of Kropff et al. (2015) for defining significantly speed-modulated neurons (see details in the revised Methods). We found that only very few identified principal neurons were significantly modulated by angular velocity (4 out of 48), which prevented statistical comparison between L2 and L3 neurons (of note, the proportion of modulated neurons did not significantly differ between L2 and L3: 2/11 and 2/25, respectively; p=0.5, Fisher’s exact test). On the other hand, the majority of FS interneurons was significantly modulated by angular velocity (13 out of 20; p<0.05). We refer to this analysis in the revised manuscript.

We have addressed this comment by providing quantitative details about the passive rotation procedures (Results) “‘Animals were head-fixed on a rotating platform and, and body-centered rotations were manually performed by the experimenter (see Video 1). Within the same recording, animals were rotated both clockwise and counterclockwise (average number of inversions, 6.6 ± 4.7; n=310 recordings) and average accelerations (1.3 ± 0.8 rad/s2), decelerations (-1.1 ± 0.7 rad/s2) and angular velocities (1.1 ± 0.4 rad/s) were within the physiological ranges reported by previous studies (…)”; For clarity, we specify that ‘animals were not actively moving during passive rotation’ (Results). We have also added one representative video of a HD cell recording (Video 1). We refer to the speed-modulation of FS INs in the Results ‘The majority of FS interneurons (13 out of 20) were significantly modulated by angular velocity (see Methods) and fired at higher rates during rotation compared to resting periods (Figure 2—figure supplement 1’).

*3) The higher theta rhythmicity in layer 2 was mostly explained by a few strongly theta modulated cells (Figure 7). Is there any evidence that these cells were different from the others? Were they all excitatory pyramidal or stellate cells or is there a chance that some were interneurons? By the way, have the authors recorded from any interneurons in the course of this study?*

To address this point, we have performed new experiments and added additional recordings from morphologically and cytochemically identified L2 and L3 neurons. The revised Figure 6 contains identified principal neurons (see [Supplementary-material SD1-data], reconstructions in Figure 6 and Figure 8, and the representative micrographs of spiny dendrites in Figure 6). The conclusions are in line with the previous manuscript, i.e. theta-rhythmicity is stronger among L2 neurons.

We agree with the Reviewers that the difference in theta-rhythmicity can be explained by few cells; in general however, and consistent with previous work (Taube 1990a, Blair and Sharp 1996, Tukker 2015) theta-rhythmic responses were very sparse among PreS neurons – principal cells as well as interneurons (Figure 6 and Figure 2—figure supplement 1). To strengthen our theta-rhythmicity finding, we performed a statistical analysis along the lines of Yartsev et al. (2011). Briefly, for each trial of the shuffling procedure, individual spike times were randomly time-shifted. For each permutation, the theta index was calculated and the procedure reiterated 1000 times. The significance value for each cell was assessed based on the resulting null distribution, i.e. a neuron was defined as significantly theta-rhythmic if the theta index was > 95th percentile of its corresponding null distribution. We found that the only significantly theta-rhythmic discharges among principal neurons were indeed contributed by L2 cells (4 out of 4). The results of this analysis are now referred to in the revised manuscript.

To address the Reviewers’ comment, we have reconstructed the morphology of the four neurons with highest theta-rhythmicity (theta-index>5; Figure 8) and two neurons with low theta- rhythmicity (theta-index<2; Figure 8) [the remaining neurons did not display complete morphology due to incomplete filling, and hence were not included in the morphological analysis].

Author response image 2.Dendritic morphologies of L2 neurons.(**A**) Reconstructed dendritic morphologies of the L2 neurons whichdisplayed theta-rhythmic spike discharges (theta- indices ≥ 5; see Figure 6 in the revised manuscript). (**B**) Reconstructed dendritic morphologies of the L2 neurons, whose spiking activitywas not rhythmically entrained by theta oscillations (theta-indices ≤ 5; see Figure 6 in the revised manuscript). (**C**) Total dendritic lengths (left bar graph) and dendritic complexity index (calculated as in Pillai et al., 2012; right bar graph) for theta-rhythmic (‘high-theta’) and non-theta -rhythmic neurons (‘low theta’) shown in **A** and **B**, respectively. Error bars represent SD. These differences were not statistically significant.**DOI:**
http://dx.doi.org/10.7554/eLife.14592.016

We can make the following observations: (i) no obvious morphological differences are apparent within these two groups (as assessed by total dendritic length and dendritic complexity index; see Figure 8), and (ii) theta-rhythmic responses were observed in both Cb+ and Cb- neurons (we note the striking similarity with MEC, where also both Cb+ and Cb- neurons can express theta-rhythmic firing, albeit with different proportions; Ray et al., 2014). It is important to specify that at present we cannot resolve structure-function relationships within L2, since our dataset of identified L2 neurons is relatively small. We specifically mention this point in the revised manuscript, and state that this remains an open question (see also response to comment 5 below).

As for interneurons; we performed new experiments, and included recordings from identified (n=3) and putative (n=17) fast-spiking (FS) interneurons in the revised manuscript. Results of the experiments are shown in Figure 2—figure supplement 1. Briefly, we show that FS interneurons can be reliably classified based on firing rate and spike width criteria, which were confirmed by cell identification (n=3 morphologically identified FS interneurons; see Figure 2—figure supplement 1). In our dataset we made the following observations: first, weakly but significantly modulated HD responses are contribute d by FS interneurons (3 out of 20; along the lines of Tukker et al., 2015), which were significantly stable between the two halves of the recording (stability analysis is referred to in the revised manuscript). Second, theta-rhythmicity was very sparse among FS interneurons. We also show one identified ‘theta cell’, which corresponded to a paravalbumin (PV) -positive interneurons; we thus extend previous observations and show that the sparse ‘theta-cells’ with the PreS (c.f.r. Taube 1990a, Blair and Sharp 1996) and are likely to correspond to a subset of PV-positive interneurons. Third, spiking activity of FS interneurons was strongly modulated by rotational movement. The results of these experiments are shown in Figure 2—figure supplement 1 and referred to in the text.

Our dataset also includes n=3 identified regular-spiking interneurons; however, given the overlap with the regular-spiking principal cell class (which is expected, given their broad AP features; see Figure 2—figure supplement 1) we do not explicitly refer to this class in the revised manuscript (directional tuning of these neurons is however shown in Figure 2—figure supplement 1).

We have addressed this comment by adding one supplementary figure (Figure 2—figure supplement 1), revising Figure 6 and performing new analysis. We refer to the theta-analysis and the firing properties of FS INs in the revised manuscript.

*4) Following up on the previous point, could the authors report more detailed descriptions of the cells they recorded within the two layers: average firing rate (partially reported in the text), peak firing rates for HD cells, waveform width, etc.*

We now provide a table which summarizes the electrophysiological properties of L2 and L3 cells ([Supplementary-material SD1-data]), and show the comparisons of spike-waveform properties, head-directionality and theta rhythmicity in the revised Figure 6. The results of this analysis are now referred to in the manuscript. We would like to point out that, following the reviewers’ suggestions, we have indeed found that L2 and L3 identified neurons display very different spike waveforms. Such differences are now shown in the revised Figure 6 and we believe they could be instrumental for future classification of tetrode-recorded units. We thank the Reviewers for this comment, as it gave us the opportunity to provide a novel insight into the electrophysiological characteristics of PreS neurons.

We have addressed this comment by performing new experiments and analysis (see revised Figure 6), adding a supplementary data ([Supplementary-material SD1-data]) and reporting average and peak firing rates of our HD cells (n=186). These data are referred to in the Results.

*5) Is there a relationship between calbindin expression and theta rhythmicity? As the authors reported that these two classes of cells have different output, it will be interesting to show that they are also functionally segregated.*

This comment is in line with the previous comment 3 (see corresponding response above). As we stated above, at present we cannot resolve whether theta-rhythmicity in PreS L2 is cell-type specific, as theta-rhythmic responses were observed in both classes (we note the striking similarity with MEC, where also both Cb+ and Cb- can express theta-rhythmic firing, albeit with different proportions; Ray et al., 2014). We quantified and compared theta-rhythmicity among Cb+ and Cb- neurons, and reported the results of this analysis in the revised manuscript (see below). We point out that in the present paper we focus on laminar differences (L2 versus L3) and that the impact of structural heterogeneity of L2 neurons remains an open question.

We have addressed this comment by performing new experiments and analysis (see revised Figure 6), the results of which are referred to in the revised manuscript (Results): ‘Theta-rhythmic spiking patterns were contributed by both calbindin-positive and calbindin-negative neurons (Figure 6), and average theta-indices did not differ significantly between the two cell classes (calbindin-positive, 2.3 ± 2.3, n=3; calbindin-negative, 3.8 ± 2.5, n=6; p=0.54; we note however that the small dataset of identified calbindin-positive neurons prevents rigorous assessment of structure-function relationships)’.

6) The experiments described were performed in head-fixed rats with passive rotation. Such an experimental design has been used before for thalamic recordings, which showed that head-direction cells can be observed under such conditions, but some differences seem to exist to free movement. Because the entire manuscript is based on this preparation, a control experiment for presubicular recordings in the set-up used here should be performed and some neurons should be recorded consecutively during passive rotation and free movement. Of course, this does not require the difficult juxta recording and labelling of the neurons, but could be done with tetrodes or silicon probes, which give more stable recordings and many cells can be recorded simultaneously. This experiment is well within the expertise of the authors and could be done relatively quickly. Importantly, it would give a quantitative measure how the same neurons fire according to head-direction in passive and free movement. This would answer if the same neurons are head-direction cells and how their head-direction tuning changes under both conditions.

We agree with the Reviewers that, although previous work indicated that HD firing is largely preserved under passive rotation, such control experiments would greatly strengthen our methodological approach and conclusions of the present work. We have thus performed a subset of juxtacellular recordings, where we have sequentially monitored the activity of the same neurons during passive rotation and free behavior [of note: we opted for juxtacellular instead of extracellular recordings because (i) they allow direct comparison with recordings of the present work, which all stem from the juxtacellular configuration and (ii) we are currently not setup for performing tetrode/silicone probe recordings].

We have employed miniaturized equipment for performing juxtacellular recordings in freely-moving animals (Tang et al., 2014), and in a subset of cases (n=4) we were able to (i) obtain a recording from a HD cell, (ii) record it during passive rotation, (iii) release the animal from the head-fixation (without losing the recording) and (iv) monitor the activity of the same HD cell during free behavior. We note this is a particularly challenging procedure, and it is the first time that juxtacellular recordings could be transferred from head-fixation to the freely-moving condition; we think that methodologically this is a potentially interesting extension of our juxtacellular procedures.

The results of these experiments demonstrate that the general tuning properties of HD cells are very similar between passive-rotation and free behavior (mean correlation coefficient of the HD tuning curves, 0.68 ± 0.20, p<0.05; n=4). We show one of these recordings in the revised Figure 2 and report correlations values in the main text. These data thus confirm that bona fide HD cells can be recorded during passive-rotation. We thank the Reviewers for this suggestion, as we think this is an important addition to our work.

To address this comment we have performed new experiments, the results of which are shown in the revised Figure 2 and referred to in the revised manuscript (Results): ‘To further confirm that bona fide HD cells can be recorded under passive rotation, in a subset of recordings (n=4) we sequentially monitored the activity of the same HD cells during head-fixation and free-behavior. To achieve this, we used miniaturized recording equipment (Tang et al., 2014), which allowed us to release the rats form head-fixation while maintaining the juxtacellular recording during free movement. As shown in the representative recording in Figure 2, the general tuning properties of the HD cells were very similar between passive-rotation and free behavior (Figure 2; mean correlation coefficient of the HD tuning curves, 0.68 ± 0.20, p<0.05; n=4)’.

7) In the Introduction the authors lay out their aim with: "Specifically, it is currently unknown how HDcells are anatomically organized within the PreS, and whether they project to MEC, as long speculated by computational models (see Giocomo et al., 2014 for review)." However, in a previous paper – the corresponding author of present manuscript was a co-auhor in this older paper – they say "..our data do indicate that at least a subset of the MEC-projecting pyramidal cells in layer 3 of the presubiculum is HD" (Tukker et al., J Neurosci 2015). A fair presentation of what has been achieved previously is necessary and should also be reflected in the significance statement.

Along with the Reviewers comment, we have changed the Introduction and Discussion accordingly.

We have addressed this comment by revising the corresponding manuscript text. Specifically, we state that ‘a previous study has indicated that HD inputs reach the MEC (Tukker et al., 2015)’ (Introduction) and that our study ‘complements earlier evidence (Tukker et al. 2015)’ (Discussion).

8) For the presented data, it is key to define the exact border between layer 2 and layer 3, which is not trivial. This should be shown with clear examples and a better description on how exactly it was decided for neurons that were located close to the border.

We agree with the Reviewers that, although in principle we are very confident about theaccuracy of our layer-assignment (which we confirmed by anatomical verification of the recording sites and by cell identification, as stated in the previous version of the manuscript) we acknowledge that there is inevitably a margin of error, especially for neurons located ‘close to the border’.

To address this point rigorously, we have performed a larger number of recording/labeling experiments and we now base all conclusions on morphologically-identified neurons (see revised Figure 6 and [Supplementary-material SD1-data]). These conclusions are in line with the previous manuscript. Our previous Figure 5, which was partially based on putatively-assigned recordings, has been removed from the revised manuscript.

We thus believe that there are no issues of layer assignment in the revised manuscript. Having the neurons identified, and a marker (calbindin) which labels PreS L2, makes it is trivial to assign neurons to the layer of origin (we note the analogy with MEC, where layer 2/3 boundary is also clearly demarcated by the calbindin staining; see previous own work e.g. Ray et al., 2014; Tang et al., 2014; Tang et al., 2015).

To address this comment, we have performed new experiments, the results of which are presented in revised Figure 6, [Supplementary-material SD1-data] and referred in the corresponding sections of the results.

*9) How was the rotation speed compared between different experiments?*

Animals were manually rotated by the experimenters (see Video 1); hence rotation was not systematically controlled across experiments. Nevertheless, we show that rotation speed, accelerations and number of inversions did not differ between L2 and L3 recordings, and hence cannot account for the (strong) electrophysiological differences between L2 and L3 neurons.

To address this comment we have performed new analysis, the results of which are referred to in the revised manuscript (Results): ‘The electrophysiological differences between L2 and L3 neurons were not accounted for by biases in rotational parameters, since average angular velocities (L2, 0.96 ± 0.31 rad/s; L3, 0.96 ± 0.33 rad/s; p=0.8), accelerations (L2, 1.41 ± 0.82 rad/s2; L3, 1.24 ± 0.89 rad/s2; p=0.4) and decelerations (L2, -1.27 ± 0.74 rad/s2; L3, 1.12 ± 0.81 rad/s2; p=0.3) were not significantly different between L2 (n=11) and L3 (n=25) recordings’.

*10) For the physiological identification of layer, a histological example of a lesion should be shown. Is it really possible to make such a small lesion to determine the border?*

This comment is in line with the comment 8 (see corresponding response above). We would like to specify that electrolytic lesions were performed in a preliminary set of experiments, were we explored the electrophysiological signatures of L2. It was sufficient for us to see that lesions were centered in (but not necessarily restricted to) L2 (as shown in Figure 9), or located at the expected distance from it (as shown in Figure 9; we also show an example of a ‘recording site’, i.e. where cell identification failed but the laminar location of the recording site could be assigned to L2). Lesions and ‘recording sites’ were thus not used for routine assignment of recordings to layers, but they were instrumental for our initial assessment of the L2 location and for subsequent targeting juxtacellular recording to this layer. The corresponding section of the Results has been clarified accordingly.

Author response image 3.Representative electrolytic lesions and ‘recording site’, which aided identification of PreS L2 in a subset of preliminary experiments.(**A**) Parasagittal section trough PreS showing the reconstructed electrode track (dotted line) and a large electrolyticlesion (dotted circle) centered on PreS L2. Green, calbindin staining. (**B**) High-m**a**gnifications view of the electrolytic lesion shown in A. (**C**) High-magnification example of another electrolytic lesion (dottedcircle and asterisk) recovered at the expecteddistance from the recording site (end of the electrode track, indicate by the arrowhead). (**D**) Parasagittal section trough PreS stained for Neurobiotin (red) and calbindin (green), showing a representative recording site’ within L. Here, cell identification by juxtacellular labeling failed; however, cell debris and small portions of dendrites (rig t panels; arrowheads) could be observed at the labeling site within PreS L2.**DOI:**
http://dx.doi.org/10.7554/eLife.14592.017

In response to this (and the above) comments, we undertook a rigorous approach: we identified a larger number of neurons and now base our conclusions on morphologically identified cells (see revised Figure 6 and [Supplementary-material SD1-data]). We have thus removed our previous Figure 5, and we do not include putatively- assigned recordings in the revised manuscript. We thus believe that there are no issues of layer assignment in the revised manuscript (see also response to comment 8 above). We thank the Reviewers for this comment, as we believe it gave us the opportunity to greatly strengthen the conclusions of our work.

We have addressed this comment by performing new experiments, the results of which are presented in the revised Figure 6. We have also removed the previous Figure 5, and L2/L3 comparisons are now based on identified neurons (see revised Figure 6). We have also clarified this issue in the revised results, where we state that electrolytic lesions were performed for assessing the location of L2 in a preliminary set of experiments (see also Methods).

*11) It would be useful to get a slightly better picture of where exactly in presubiculum the study was done, perhaps with a zoomed-out anatomical illustration, and location on an atlas, so that the exact region is more obvious for anyone who might want to pursue this further.*

We provide a better overview and low-magnification pictures of the dorsal PreS (where thisstudy was performed; Figure 1—figure supplement 1) along with a number of molecular and histochemical markers which outline the PreS borders (Figure 1—figure supplement 2).

We have addressed this comment by including 2 supplementary figures (Figure 1—figure supplement 1 and Figure 1—figure supplement 2). The results of these experiments are referred to in the revised text (Results).

*The analysis is focused on MEC but the interconnections with RSC are also very important to understand, as the authors note in the discussion, and the impact statement could be adapted to include this. We would like to know more about which part of RSC (e.g. AP location, layer etc) was targeted and exactly what the labelling patterns were.*

We provide now a better overview of which portion of RS cortex was targeted for retrograde tracer injections. First, we show how this region and the corresponding border with PreS can be reliably outlined by a number of markers and histochemical stainings. Second, we show representative injection sites. Third, we show the corresponding labeling pattern within the contralateral PreS and a quantification of retrogradely-labelled neurons across PreS layers. These results are shown in Figure 1—figure supplement 2 and figure supplement 3 and referred to in the text.

We have addressed this comment by including 2 supplementary figures (Figure 1—figure supplement 2 and Figure 1—figure supplement 3). The results of these experiments are referred to in the revised text (Results) and the revised Methods.

We also need a lot more details about the behavioural manipulation – how was the animal rotated, with what angular acceleration and velocity, how many reversals, what visual cues were available, etc.

We provide these details in the revised manuscript (Results and Methods, see below). For clarity, we also show a video of a representative HD cell (Video 1). We have also performed additional analysis, and provide average values of angular acceleration and deceleration, angular velocity and number of reversals (see below). As for the available visual cues, we specify that bot habituation to head fixation/rotation and experiments were performed in the ‘cue-rich’ environment of the laboratory setting. Under dim illumination, rats had thus visual access to both proximal cues (e.g. computer screens, cold-light source, stereomicroscope) and distal cues (e.g. Faraday cage, ceiling, curtains), including the experimenter, which was always located in the same relative position during the passive rotation experiment. These details are now referred to in the text.

We have addressed this comment by performing new analysis, the results of which are referred to in the revised manuscript (Results): ‘Animals were head-fixed on a rotating platform and, and body-centered rotations were manually performed by the experimenter (see Video 1). Within the same recording, animals were rotated both clockwise and counterclockwise (average number of inversions, 6.6 ± 4.7; n=310 recordings) and average accelerations (1.3 ± 0.8 rad/s2), decelerations (-1.1 ± 0.7 rad/s2) and angular velocities (1.1 ± 0.4 rad/s) were within the physiological ranges reported by previous studies’. We also provide a video of a representative HD cell (Video 1) and more details about the available cues in the revised Methods: ‘Habituation and recordings were performed under slightly-dimmed ambient illumination in the ‘cue-rich’ environment of the laboratory setting. Thus both during habituation and recordings, rats had visual access to both proximal cues available in the immediate vicinity (e.g. computer screens, cold-light source, stereomicroscope, etc…) and distal cues (i.e. Faraday cage, ceiling, curtains, etc…), including the experimenter, which was always located in the same relative position during the passive rotation experiment. These cues were thus the most likely source of ‘anchoring’ stability to HD firing (see e.g. Knierim et al., 1995 for review)’.

[Editors' note: further revisions were requested prior to acceptance, as described below.]

*Essential revisions:*

1) The additional data and figures are particularly useful. Please include one of the sub-parts of Figure 1—figure supplement 1 (micrograph plus line drawing) in the main paper, to help the reader visualize the anatomy.

In line with the Reviewers’ comment, we have added one panel to our revised Figure 1 (new Figure 1).

*2) Please complete the presentation with a wiring diagram including the cell types, their properties, distribution and connections, summarizing the take-home message visually (akin to a graphical abstract).*

We have made such diagram. This diagram could be included as Figure 6—figure supplement 1 (we explored alternative options –e.g. adding it to our current Figure 6- but we find it difficult to include it as part of existing figures without significantly compromising their size/resolution).

However, we feel that this schematic representation is rather redundant and adds too little information for justifying its inclusion as a stand-alone figure. Moreover, it will increase our already-large number of supplementary items. We would therefore prefer not to include the diagram in the revised manuscript. However, if the Reviewers/Editor feel it is necessary, we will include it and upload it as Figure 6—figure supplement 1.

[Editors' note: further revisions were requested prior to acceptance, as described below.]

*The manuscript has been improved but there are some remaining issues that need to be addressed before acceptance, as outlined below:*

*The reviewers found the article adequately revised but one reviewer had the following recommendations. The reviewer found the additional data and figures particularly useful and recommended that one of the sub-parts of Figure 1—figure supplement 1 (micrograph plus line drawing) be moved to the main paper to help the readers visualize the anatomy. Also this reviewer suggested that the authors add a wiring diagram including the cell types, their properties, distribution and connections, summarizing the take-home message visually (akin to a graphical abstract).*

In line with the Reviewers’ comment, we have added one panel to our revised Figure 1 (new Figure 1).

We have uploaded the diagram as Figure 6—figure supplement 1